# Multivalent Tau/PSD-95 interactions arrest in vitro condensates and clusters mimicking the postsynaptic density

Zheng Shen [1], Daxiao Sun[2], Adriana Savastano [1], Sára Joana Varga[1], Maria-Sol Cima-Omori [1], Stefan Becker [3], Alf Honigmann [2,4] & Markus Zweckstetter [1,3] ✉

Alzheimer's disease begins with mild memory loss and slowly destroys memory and thinking. Cognitive impairment in Alzheimer's disease has been associated with the localization of the microtubule-associated protein Tau at the postsynapse. However, the correlation between Tau at the postsynapse and synaptic dysfunction remains unclear. Here, we show that Tau arrests liquid-like droplets formed by the four postsynaptic density proteins PSD-95, GKAP, Shank, Homer in solution, as well as NMDA (*N*-methyl-D-aspartate)-receptor-associated protein clusters on synthetic membranes. Tau-mediated condensate/cluster arrest critically depends on the binding of multiple inter-action motifs of Tau to a canonical GMP-binding pocket in the guanylate kinase domain of PSD-95. We further reveal that competitive binding of a high-affinity phosphorylated peptide to PSD-95 rescues the diffusional dynamics of an NMDA truncated construct, which contains the last five amino acids of the NMDA receptor subunit NR2B fused to the C-terminus of the tetrameric GCN4 coiled-coil domain, in postsynaptic density-like condensates/clusters. Taken together, our findings propose a molecular mechanism where Tau modulates the dynamic properties of the postsynaptic density.

Cognitive impairment of Alzheimer's disease patients, in particular in the early stages of the disease, is caused by synaptic dysfunction[1]. Synaptic dysfunction and perturbed neural network activity may also be responsible for the increased incidence of seizures in Alzheimer's disease patients[2]. Synaptic dysfunction and perturbed neural network activity arise from the excessive activity of the ionotropic glutamate receptor NMDA (*N*-methyl-D-aspartic acid)[3]. Overactivation of NMDA receptors in Alzheimer's disease has been associated with the presence of the microtubule-associated protein Tau at the postsynapse[4–8]. Localization of Tau at the postsynapse is promoted by phosphorylation of Tau[9–12]. The mechanistic correlation between the postsynaptic accumulation of Tau and synaptic dysfunction in Alzheimer's disease however remains unclear.

Biomolecular condensates have been implicated in a wide range of cellular activities[13–17]. In electron micrographs of excitatory synapses, an electron-dense molecular cluster is adjacent to the postsynaptic plasma membrane, known as the postsynaptic density (PSD)[18,19]. It comprises hundreds of signaling and scaffolding proteins connected to transmembrane receptors and is important for synaptic plasticity[19]. The PSD displays features associated with biomolecular condensates, such as high protein concentrations, fast growth and shrinkage, mobility of its components and rapid component exchange with the dilute cytoplasm of dendritic spines[20]. In support of the description of the PSD as biomolecular condensate, selected PSD proteins undergo liquid-liquid phase separation upon mixing[20–22]. In particular, the mixing of four important PSD scaffold proteins, the postsynaptic

[1]German Center for Neurodegenerative Diseases (DZNE), Von-Siebold-Str. 3a, 37075 Göttingen, Germany. [2]Max Planck Institute of Molecular Cell Biology and Genetics, Dresden, Germany. [3]Max Planck Institute for Multidisciplinary Sciences, Department of NMR-based Structural Biology, Am Fassberg 11, 37077 Göttingen, Germany. [4]Technische Universität Dresden, Biotechnologisches Zentrum (BIOTEC), Dresden, Germany. ✉e-mail: Markus.Zweckstetter@dzne.de

density protein-95 (PSD-95), guanylate kinase (GK)-associated protein (GKAP), SH3 and multiple ankyrin repeat domain protein (Shank), as well as the Shank-interacting protein Homer, which connect the receptors and ion channels of the postsynaptic membrane to the actin cytoskeleton[23–25], results in the formation of liquid-like droplets in solution and NMDA-receptor-associated protein clusters on membranes[21]. PSD-95/GKAP/Shank/Homer-based PSD condensates may thus serve as a molecular platform to study regulatory mechanisms of synaptic maturation and plasticity[21].

Tau is located throughout the neuron and regulates the dynamics and organization of microtubules[26,27]. Additionally, Tau can be found at the postsynapse, where it might interact with PSD-95[5,28]. Different Tau species (oligomeric, misfolded and phosphorylated) colocalize with PSD-95 at the postsynaptic site in human postmortem samples[29]. Consistent with these findings, colocalization of phosphorylated Tau and PSD-95 increases during Alzheimer's disease progression across clinically stratified groups (normal, mild cognitive impairment, Alzheimer's disease) in the frontal cortex[30]. Additionally, signals of AT8, AT100, and AT180 antibodies, which recognize phosphoepitopes that are abundant in pathological aggregated and filamentous Tau of Alzheimer's disease, are elevated in Tau transgenic mice, in particular in PSD fractions[31]. The interaction of Tau with PSD-95 may lead to over-activation of NMDA receptors resulting in excitotoxicity[5,28]. Consistently, clearance of postsynaptic Tau attenuated tauopathy and improved cognitive behavior in tauopathy mouse models[32], and Tau reduction protected against excitotoxicity and seizures[33,34]. However, little is known about the molecular pathways linking postsynaptic Tau and synaptic dysfunction.

The PSD is a dynamic entity and continuous remodeling of the PSD is critical for synaptic activity[35,36]. This suggests that changes in the dynamic properties of the PSD may affect the mobility of PSD-95 and other PSD proteins and thus affect the organization and clustering of NMDA receptors. Altered organization/clustering of NMDA receptors may result in receptor overactivation, excitotoxicity and synaptic dysfunction[36,37].

We therefore investigate in the current work whether Tau concentrates in in vitro condensates/clusters that mimic the PSD, changes their material properties and affects the diffusional dynamics of an NMDA truncated construct, which contains the last five amino acids of the NMDA receptor subunit NR2B fused to the C-terminus of the tetrameric GCN4 coiled-coil domain. Our findings suggest a mechanism for prolonged clustering and activity of NMDA receptors on the postsynaptic membrane and thus for excitotoxicity-associated cognitive dysfunction. In addition, we show that Tau-mediated changes of in vitro condensate/cluster dynamics can be reversed by a phosphorylated peptide that binds with high affinity to PSD-95.

## Results

### Tau reduces the dynamics of phase-separated droplets formed by PSD proteins

To investigate the potential role of Tau at the postsynapse, we established the previously developed in vitro platform, which is based on the liquid-liquid phase separation of PSD proteins and enables mechanistic studies of synaptic maturation and plasticity (Fig. 1A)[21]. To this end, we mixed the four PSD scaffold proteins PSD-95, GKAP, Shank and Homer (PSD 4X) at a 1:1:1:1 stoichiometry[21]. Mixing of the four proteins resulted in immediate phase separation, in agreement with previous observations (Fig. 1B, upper part)[21]. We then mixed the four PSD proteins and full-length human 2N4R Tau (441 residues; further referred to as Tau) at an equal molar ratio in the same buffer and observed the formation of spherical droplets (Fig. 1B, bottom part). Fluorescence microscopy demonstrated that Tau is recruited into the PSD droplets (Fig. 1B, C). We also observed Tau enrichment in PSD-95/GKAP condensates, indicating that Shank and Homer are not required for Tau recruitment into the PSD condensate (Supplementary Fig. 1A).

The dynamic nature of the PSD is critical for synaptic activity[35,36]. To investigate whether Tau recruitment changes the dynamic properties of the PSD condensate, we photobleached PSD-95 inside PSD condensates without and with Tau. We observed fast fluorescence recovery of PSD-95 in the absence of Tau (Fig. 1D, E)[21]. In contrast, the recovery of the PSD-95 fluorescence was slowed with Tau in the droplets (Fig. 1D, E). In addition, Tau decreased the mobile fraction of PSD-95 (Fig. 1F). The dynamic arrest (for a definition of dynamic arrest, please see ref. 16) of PSD-95 was even stronger in PSD-95/GKAP/Tau droplets (Supplementary Fig. 1B–D). Collectively, Tau enrichment reduces the fluidity of the PSD condensate.

As PSD-95 binds to NMDA receptors, we next tested whether the Tau-mediated dynamic arrest of PSD-95 modulates NMDA receptor dynamics. We began by generating a tetramethylrhodamine (TMR)-labeled NMDA truncated construct, which contains the last five amino acids of the NMDA receptor subunit NR2B fused to the C-terminus of the tetrameric GCN4 coiled-coil domain (design based on ref. 21 called TMR-NR2B hereafter). TMR-NR2B contains the binding site for PSD-95 and mimics the tetrameric structure of NMDA receptors. Fluorescence microscopy demonstrated the recruitment of TMR-NR2B into PSD-95/GKAP/Shank/Homer droplets (Fig. 1G). We then characterized the mobility of TMR-NR2B within PSD/NR2B droplets in the presence of increasing Tau concentrations (Fig. 1G). FRAP analysis showed that Tau reduced the fluorescence recovery of TMR-NR2B (Fig. 1H). Tau also decreased TMR-NR2B mobile fractions from 79% (without Tau) to 68% and 66% at 0.5 μM and 1 μM Tau, respectively (Fig. 1I). The data demonstrate that Tau decreases TMR-NR2B dynamics already below cellular Tau concentrations (i.e., ~1–2 μM; ref. 38).

### Tau partitions into membrane-associated NR2B/PSD clusters

PSD assembly occurs at the postsynaptic membrane to bind and cluster membrane-bound NMDA receptors[18]. To complement our solution experiments with NMDA receptors bound to lipid membranes, we used a previously established reconstituted membrane system[21]. We prepared an N-terminal His$_6$-tagged GCN4-NR2B construct (called His-NR2B hereafter) and attached it to supported lipid bilayers consisting of 4% of the headgroup-modified lipid chelator Ni$^{2+}$-NTA-DGS (Fig. 2A). Fluorescence microscopy showed that membrane-anchored His-NR2B is homogeneously distributed and freely diffusible in the lipid bilayer (Supplementary Fig. 2A–C)[21]. The addition of Tau alone did not change the distribution and dynamics of His-NR2B on the membrane (Supplementary Fig. 2A–C). In contrast, upon the addition of premixed PSD 4X scaffold proteins, His-NR2B progressively coalesced into larger micron-sized clusters (Supplementary Fig. 2D).

We then performed the membrane-based NR2B clustering assay in the presence of Tau. His-NR2B rapidly clustered within 15 min after the addition of soluble PSD components (Fig. 2B). In addition, PSD-95 fluorescence overlaid with His-NR2B fluorescence, indicating that PSD-95 is present in the clusters (Fig. 2B). Fluorescence microscopy further demonstrated that the Tau protein is recruited and enriched in the His-NR2B clusters (Fig. 2B).

Next, we monitored the Tau-mediated impact on the dynamic properties of the membrane-associated NR2B/PSD clusters. Fluorescence recovery after photobleaching showed that the mobility of the membrane-anchored His-NR2B is low in NR2B/PSD membrane clusters (Supplementary Fig. 2E–G). With less than 20% of His-NR2B being mobile, it is challenging to distinguish between His-NR2B dynamics without and with Tau (Supplementary Fig. 2E–G). We therefore focused on the dynamics of PSD-95 in the NR2B/PSD membrane clusters. We bleached fluorescently labeled PSD-95 and monitored fluorescence recovery. The PSD-95 fluorescence almost fully recovered within 5 min (Fig. 2C). Presence of Tau in the membrane-

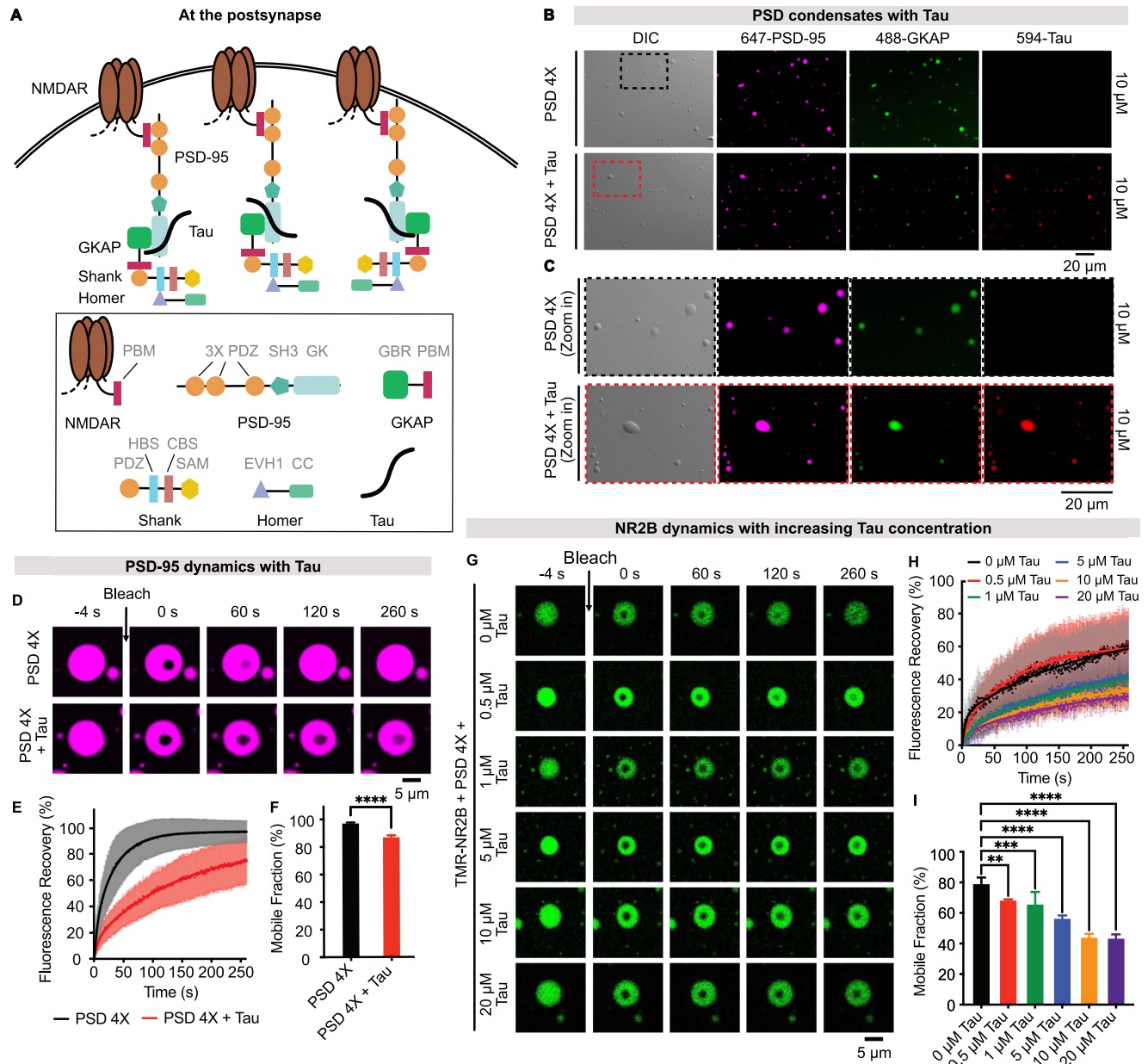

**Fig. 1 | Tau arrests PSD condensates and decreases the mobility of NMDA receptors. A** Schematic diagram illustrating the major protein components of PSD condensates: NMDA receptors (NMDARs), PSD-95, GKAP, Shank, and Homer. The domain organization of the proteins is shown below. Tau is shown in black. **B**, **C** DIC and fluorescence microscopy of PSD condensates containing four PSD scaffold proteins (PSD-95/GKAP/Shank/Homer) without (top; "PSD 4X") and with Tau (bottom; "PSD 4X + Tau"). PSD-95 and Tau were labeled with Alexa 647 and Alexa 594, respectively. The concentration of each protein was 10 μM. Ten and nine measurements were done for "PSD 4X" and "PSD 4X + Tau", respectively. **D** Representative FRAP images of PSD-95 in PSD droplets without and with Tau. **E** Quantification of PSD-95 dynamics in PSD droplets without (black) and with Tau (red). Data are presented as mean values ± SD from $n$ independent experiments. For "PSD 4X" condition, $n = 9$; for "PSD 4X + Tau" condition, $n = 8$. FRAP curves (dots) were fitted with a bi-exponential function (solid lines). **F** Mobile fractions of PSD-95 were derived from the fitted averaged FRAP curves in (**E**), which were averaged

from 9 and 8 independent experiments for "PSD 4X" and "PSD 4X + Tau" conditions, respectively. Error bars represent the standard deviation of curve fits; unpaired and two-tailed $t$-test with Welch's correction: ****$p \le 0.0001$. **G** Representative FRAP images of TMR-labeled NR2B (TMR-NR2B; 1 μM) in NR2B/PSD droplets with increasing concentrations of Tau. The concentration of the four PSD scaffold proteins was 20 μM. **H** Quantification of TMR-NR2B dynamics. Data are presented as mean values ± SD from $n$ independent experiments. The number of experiments ($n$) for each condition is as follows: "0 μM Tau" ($n = 5$), "0.5 μM Tau" ($n = 6$), "1 μM Tau" ($n = 5$), "5 μM Tau" ($n = 4$), "10 μM Tau" ($n = 4$), and "20 μM Tau" ($n = 5$). **I** Mobile fractions of TMR-NR2B were derived from the fitted averaged FRAP curves, which were averaged from $n$ independent experiments. For each condition, $n$ corresponds to the number of independent FRAP experiments indicated in (**H**). Error bars represent the standard deviation of curve fits; one-way ANOVA: **$p \le 0.0021$, ***$p \le 0.0002$, ****$p \le 0.0001$. Source data are provided as a source data file.

associated PSD clusters, however, decreased the PSD-95 dynamics (Fig. 2C, D). The mobile fraction of PSD-95 dropped from 93% (without Tau) to 69%, 64%, and 59% upon the addition of 1 μM, 2 μM, and 4 μM Tau, respectively (Fig. 2E), showing that partitioning of Tau decreases the dynamics of the membrane-attached PSD scaffold.

## Tau engages in multivalent interactions with PSD-95

Multivalent protein interactions drive phase separation and are important for condensate formation of PSD-95 as well as Tau[21,39–41]. To gain insight into the molecular determinants of the Tau-induced arrest of PSD condensate dynamics, we prepared [15]N-labeled 2N4R Tau and

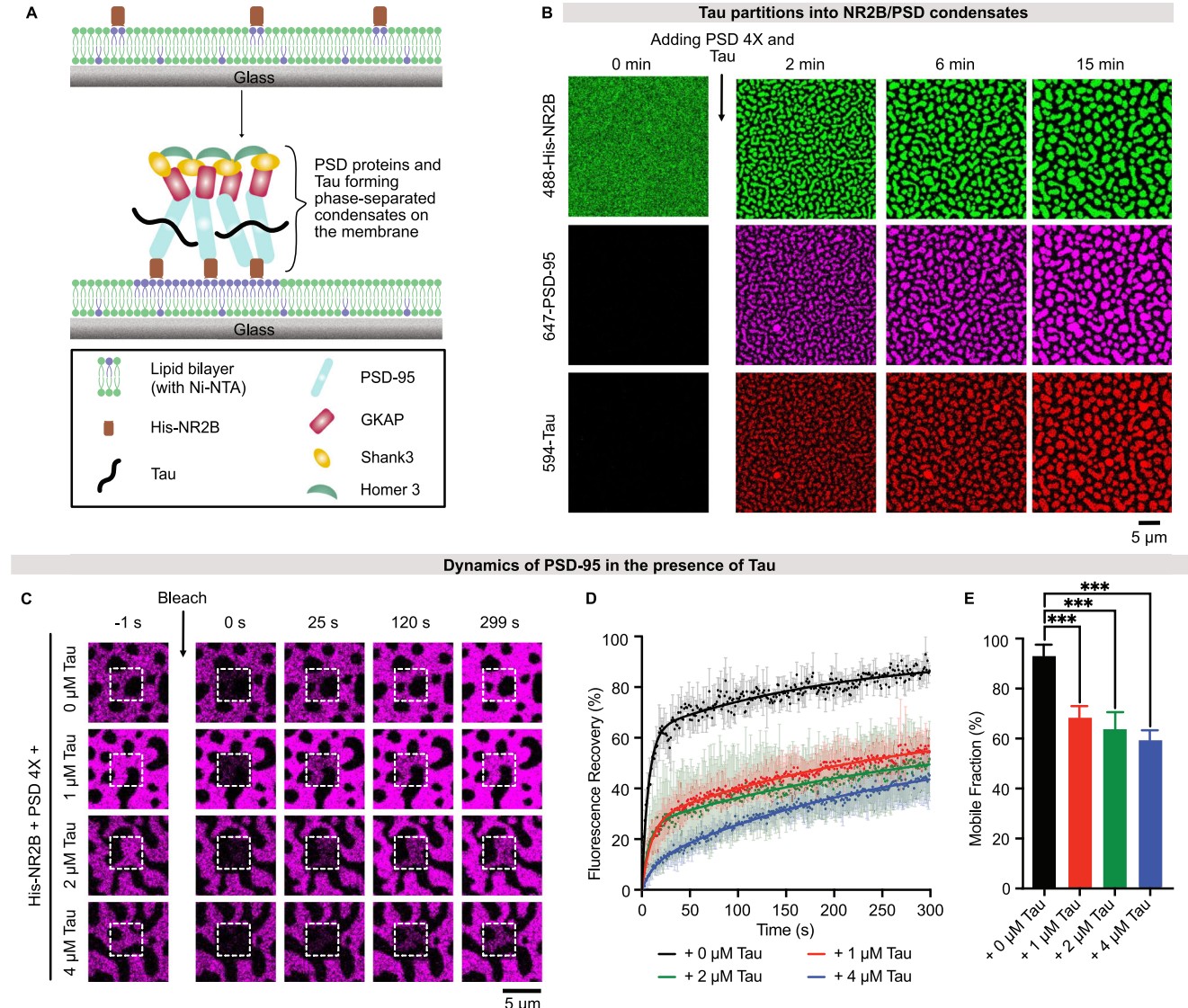

**Fig. 2 | Tau partitions into membrane-anchored NR2B/PSD clusters and decreases PSD-95 dynamics. A** Schematic illustration of Tau recruitment into TMR-NR2B/PSD clusters on supported membranes. **B** Time-lapse confocal images of Alexa 488-labeled His-NR2B clustering after the addition of PSD 4X scaffold proteins and Tau (each at 1 μM). **C** Representative FRAP images of PSD-95 in membrane-anchored TMR-NR2B/PSD clusters with increasing concentration of Tau. The bulk concentration of each PSD scaffold protein was 1 μM. **D** FRAP-based quantification of PSD-95 dynamics in membrane-anchored PSD clusters in the presence of increasing Tau concentrations. Each FRAP curve represents the average of three FRAP measurements. **E** Mobile fractions of PSD-95 in membrane-anchored PSD clusters were derived from the fitted averaged FRAP curves, which were averaged from three independent experiments for each condition. Error bars represent the standard deviation of curve fits; one-way ANOVA: ***$p ≤ 0.0002$. Source data are provided as a source data file.

monitored PSD-95 binding to Tau using two-dimensional $^{1}$H-$^{15}$N correlation spectra (Fig. 3A, B). The addition of a 6-fold molar excess of PSD-95 attenuated the intensities of selected Tau cross-peaks (Fig. 3B). As peak positions also shifted slightly, the Tau/PSD-95 interaction is in an intermediate exchange regime. The strongest PSD-95-induced Tau peak intensity attenuations were observed in the microtubule-binding repeats R1 to R3, in particular for residues Q244 to N255 (R1 repeat), V275 to S285 (R2), and G304 to D314 (R2/R3) (Fig. 3C). Additional peak intensity attenuation occurred at the N-terminus and in the C-terminal domain (Fig. 3C). Several domains of Tau thus engage in Tau/PSD-95 interactions. In contrast, no interaction between Tau and any of the three other PSD scaffold proteins (GKAP, Shank, Homer) was detected by NMR spectroscopy (Supplementary Fig. 3).

To identify the interaction sites of Tau in PSD-95, we recombinantly produced a truncated construct of PSD-95 containing the SH3 and guanylate kinase (GK) domain (called PSD-95_SH3-GK hereafter)

(Fig. 3A). We then performed NMR-based binding studies of Tau with PSD-95_SH3-GK (Fig. 3B, C). Highly similar interaction profiles were observed between Tau/PSD-95_SH3GK and Tau/PSD-95 (Fig. 3C). The SH3 and GK domain of PSD-95 thus fully represent the multivalent Tau/PSD-95 interaction, in agreement with a previous Tau/PSD-95 interaction study in cultured cells[28].

We then asked which parts of PSD-95_SH3-GK interact with Tau. We selected the Tau fragments Q244-N255 and G304-D314 (called Tau_R1 and Tau_R3, respectively), which displayed the strongest NMR signal attenuation (Fig. 3C) and predicted their complex structures with PSD-95_SH3-GK using the neural network-based structure prediction software AlphaFold2[42]. According to the AlphaFold2-predicted 3D structures, the GK domain of PSD-95 interacts with Tau via a binding pocket formed by the GMP-binding domain (GMP-BD) and the CORE/LID domain (Fig. 3D). This binding pocket was previously shown to interact with the rationally designed peptides GKI-QSF, DLS and

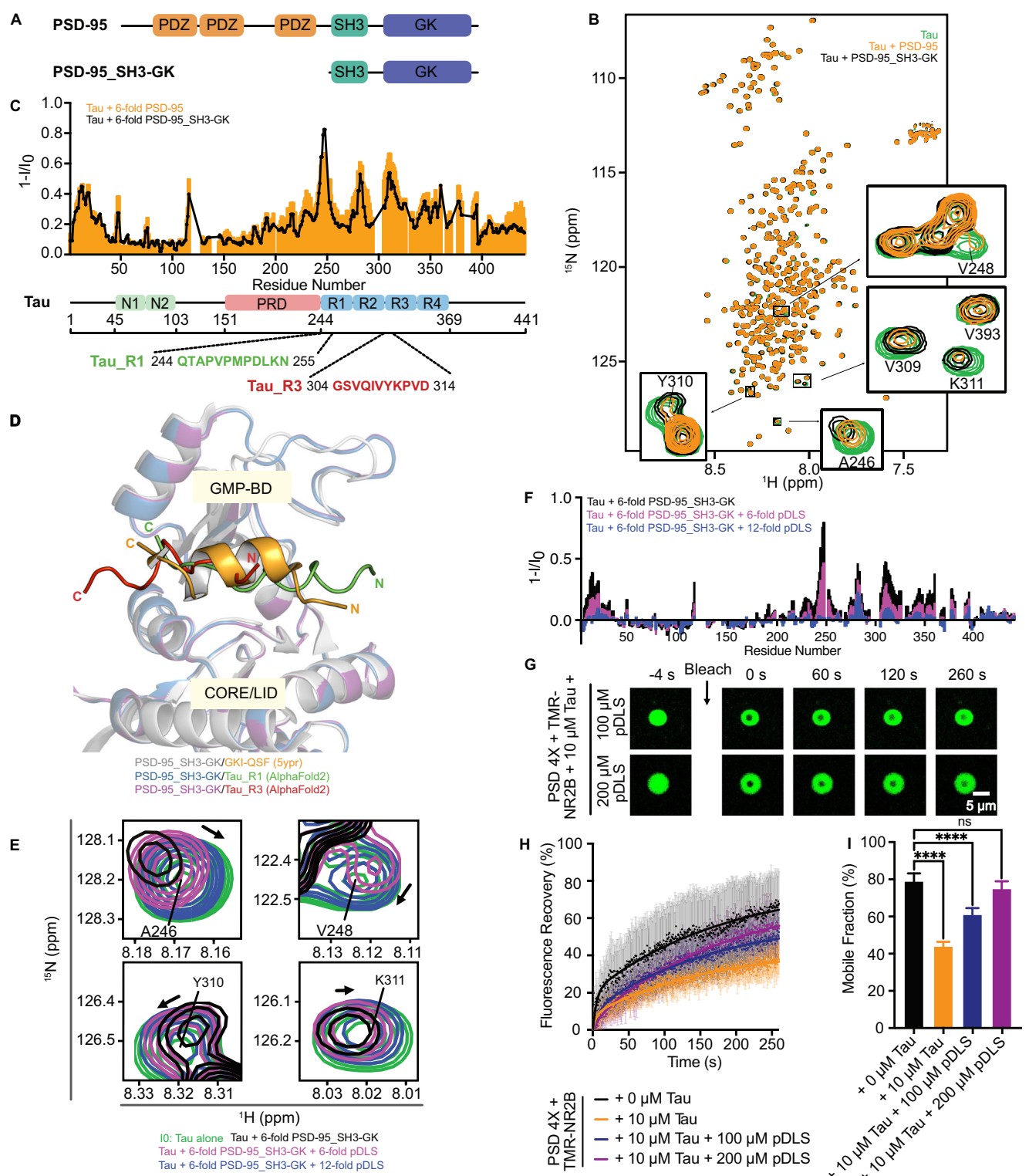

pDLS[43]. For the DLS peptide, it was further shown that its binding to the GMP-binding domain inhibits excitatory synaptic activity[43]. X-ray crystallography revealed that the rationally designed peptides fold into an α-helix upon binding to the GK domain of PSD-95 (Fig. 3D)[43]. Similar to Map1a[44], however, Tau is predicted to bind to this binding pocket in an extended conformation (Fig. 3D), potentially due to the presence of one (Tau_R3) or three proline residues (Tau_R1) (Fig. 3C).

To validate the predicted Tau-binding pocket, we performed NMR-based competition experiments with the pDLS peptide. The pDLS peptide was selected because it has much higher affinity

($K_d = 2.3 \pm 0.8$ nM) to the PSD-95 GK domain than the GKI-QSF ($K_d = 1.3 \pm 0.8$ μM) and DLS ($K_d = 0.7 \pm 0.1$ μM) peptides[43]. Upon addition of pDLS, Tau peaks previously attenuated by PSD-95_SH3-GK, including A246, V248, Y310, and K311, were recovered (Fig. 3E). In addition, the slight PSD-95_SH3-GK-induced changes in peak positions were reversed such that at 2-fold excess of pDLS over PSD-95_SH3-GK the cross-peaks were highly similar to those of Tau alone (Fig. 3E). Residue-specific analysis confirmed the competition of pDLS with Tau for binding to the GK domain of PSD-95 (Fig. 3F). In addition, it showed that not only the PSD-95-induced peak intensity attenuation in Tau's

**Fig. 3 | Multivalent interactions between Tau and the GK Domain of PSD-95.**
**A** Domain architecture of PSD-95. The PSD-95_SH3-GK construct is shown below.
**B** Superposition of $^1H$-$^{15}N$ HSQC spectra of 20 μM Tau alone (green), upon the addition of 6-fold molar excess (120 μM) of either full-length PSD-95 (orange) or PSD-95_SH3-GK (black). Selected cross-peaks of Tau significantly attenuated by PSD-95 or PSD-95_SH3-GK are highlighted. **C** NMR signal intensity profiles corresponding to (**B**) displaying the signal intensity changes of Tau HSQC peaks upon the addition of PSD-95 (orange bars) and PSD-95_SH3-GK (black lines). $I_0$ are signal intensities of Tau alone. The domain organization of Tau is shown below the intensity plots; N1 and N2 stand for the N-terminal inserts; PRD stands for proline-rich domain; R1, R2, R3, and R4 represent the microtubule-binding repeats of Tau. **D** Superposition of PSD-95_SH3-GK structures bound to GKI-QSF (yellow; PDB code: 5ypr), Tau_R1 (green; AlphaFold2 prediction) and Tau_R3 (red; AlphaFold2 prediction). GMP-BD and CORE/LID domains of PSD-95 are indicated. **E** Superposition of selected cross-peaks from $^1H$-$^{15}N$ HSQC spectra of Tau alone (green), in the presence of 6-fold molar excess of PSD-95_SH3-GK (black) and PSD-95_SH3GK/pDLS

complexes using Tau: PSD-95_SH3GK: pDLS molar ratios of 1:6:6 (magenta) and 1:6:12 (blue). Black arrows indicate peak shifts upon pDLS addition. **F** NMR signal intensity profiles corresponding to (**E**) displaying signal intensity changes of Tau HSQC peaks upon the addition of PSD-95_SH3-GK and PSD-95_SH3-GK/pDLS complexes. Same color codes as (**E**). $I_0$ are signal intensities of Tau alone. **G** Representative FRAP images of TMR-NR2B in NR2B/PSD/Tau droplets with increasing concentration of pDLS. 1 μM TMR-NR2B, 20 μM of each PSD scaffold protein, and 10 μM Tau were added. **H** Quantification of TMR-NR2B dynamics. Data are presented as mean values ± SD from $n$ independent experiments. The number of experiments ($n$) for each condition is as follows: "+ 0 μM Tau" ($n = 5$), "+ 10 μM Tau" ($n = 4$), "+ 10 μM Tau + 100 μM pDLS" ($n = 4$), and "+ 10 μM Tau + 200 μM pDLS" ($n = 3$). Fitted FRAP curves are shown as solid lines. The corresponding FRAP images of TMR-NR2B dynamics in NR2B/PSD droplets without and with 10 μM Tau are shown in Fig. 1G. Indent FRAP experiments are indicated in (**H**). Error bars represent the standard deviation of curve fits; one-way ANOVA: ****$p \le 0.0001$. Source data are provided as a source data file.

repeat region is reversed, but all other domains of Tau are unable to bind to PSD-95 in the presence of pDLS. The combined data demonstrate that multiple interaction motifs of Tau bind to the GMP-BD/CORE/LID pocket of the GK domain of PSD-95.

### Inhibition of Tau-mediated dynamic arrest of PSD condensates
Next, we investigated whether pDLS is able to reverse the Tau-induced decrease in NR2B/PSD condensate dynamics. We added pDLS to preformed PSD/NR2B/Tau droplets (Fig. 3G). Large excess (100 or 200 μM) of pDLS over Tau (10 μM) was used because Tau and PSD-95 are highly concentrated in the PSD droplets (Fig. 1), while the condensate-enrichment of pDLS is unknown.

Fluorescence recovery curves showed that the fluorescently labeled NR2B recovered faster and more fully in the presence of pDLS (Fig. 3H). Mobile fractions of TMR-NR2B in the PSD/NR2B/Tau droplets without and with 100 or 200 μM pDLS were 44%, 61%, and 75%, respectively (Fig. 3I). We observed a close similarity between the mobile fractions of TMR-NR2B in condensates with 200 μM pDLS (75%) and without Tau/pDLS (79%). The inhibitory peptide pDLS thus almost fully reversed the Tau-induced dynamic arrest of the PSD condensate (Fig. 3I). The combined data highlight the importance and specificity of Tau/PSD-95 multivalent interactions on the Tau-mediated dynamic arrest of the PSD condensate, which can be overturned by competitive inhibitory peptides.

### Phosphorylated Tau decreases PSD condensate dynamics
Tau can be phosphorylated at its tyrosine residues by the tyrosine kinase Fyn (named pTau$^{Fyn}$). pTau$^{Fyn}$ may induce excessive activity of NMDA receptors[8]. To gain insight into the potential activity of pTau$^{Fyn}$, we phosphorylated full-length 2N4R Tau in vitro using Fyn. NMR chemical shift perturbation identified phosphorylated tyrosine residues, including the disease-associated phosphorylated Y18 (pY18; Fig. 4A)[45]. DIC and fluorescence microscopy revealed droplet formation when the four PSD scaffold proteins and pTau$^{Fyn}$ were mixed at an equal molar ratio (Fig. 4B, C), and pTau$^{Fyn}$ was recruited into membrane-anchored PSD/NR2B clusters (Fig. 4D). More and larger PSD droplets were formed with pTau$^{Fyn}$ when compared to unmodified Tau or the four PSD scaffold proteins alone (Fig. 4E, F). We further observed that pTau$^{Fyn}$ reduced PSD-95 dynamics more than unmodified Tau (Fig. 4G, H): 87% and 60% of PSD-95 were mobile species in PSD condensates with unmodified Tau and with pTau$^{Fyn}$, respectively (Fig. 4I).

To investigate the molecular basis of the pTau$^{Fyn}$-mediated dynamic arrest of the PSD condensate, we studied the pTau$^{Fyn}$/PSD-95 interaction using NMR spectroscopy. NMR intensity analysis of pTau$^{Fyn}$ showed a similar interaction profile with PSD-95_SH3-GK as unmodified Tau (Fig. 4J). However, the intensities of pTau$^{Fyn}$ residues in the R3 region were less attenuated than in the case of unmodified Tau

(Fig. 4J). This modulation appears to be small and is only detected for Fyn-phosphorylated Tau, but not for c-Abl phosphorylated Tau or a Tau mutant protein in which Y394 was substituted by asparagine (Tau_Y394N) (Supplementary Fig. 4). The data demonstrate that both unmodified and phosphorylated Tau partition into in vitro condensates containing PSD-95 and change their diffusional properties.

## Discussion
Tau protein is linked to the pathogenesis of Alzheimer's disease and other neurodegenerative disorders[1]. Additionally, it has been associated with disease-associated neural network dysfunction and cognitive impairment of patients[1,3]. To develop treatment options, it is therefore important to understand the molecular mechanisms of Tau at the postsynapse. Using biochemical reconstitution, we showed that Tau concentrates inside in vitro condensates and membrane-associated clusters, which are formed by major PSD scaffold proteins and were previously suggested to mimic specific aspects of the postsynaptic density. We further showed that the recruitment of Tau perturbs the dynamic properties of these condensates/clusters both in solution and anchored to a lipid bilayer. Additionally, the Tau-mediated arrest decreased the solution condensate dynamics of a truncated NMDA construct, which can bind to PSD-95[21,46]. We found this process to depend on multivalent interactions between Tau and the GK domain of PSD-95 (Fig. 5).

We demonstrated that Tau concentrates in PSD condensates and membrane-anchored clusters and decreases the mobility of PSD-95. Our results propose a relay mechanism in which multivalent interactions between the intrinsically disordered protein Tau and the GK domain of PSD-95 cause cross-linking of PSD-95 molecules, leading to dynamic arrest of PSD condensates and clusters and thus indirectly to decreased NMDA receptor mobility through PSD-95 and NMDA receptor interaction. The Tau-induced decrease in NMDA receptor mobility may influence the clustering and trafficking of NMDA receptors in and into dendritic spines, which may contribute to synaptic dysfunction in Alzheimer's disease (Fig. 5). While changes in the material properties of biomolecular condensates were previously suggested to play a role in neurodegenerative diseases[14,16,47,48], our work provides evidence for the impact of condensate arrest on a receptor that is critical for synaptic function and neural plasticity.

The postsynaptic density is involved in multiple postsynaptic functions, including cell-cell adhesion, regulation of receptor clustering and modulation of receptor function[19]. The Tau-induced dynamic arrest of condensates and clusters mimicking the PSD may therefore have additional PSD-dependent consequences beyond changing NMDA receptor dynamics and thus connect different previously reported Tau-related pathomechanisms. For example, previous work showed that acetylated Tau disrupts synaptic signaling that is mediated by the PSD scaffolding protein KIBRA[49,50]. Additionally,

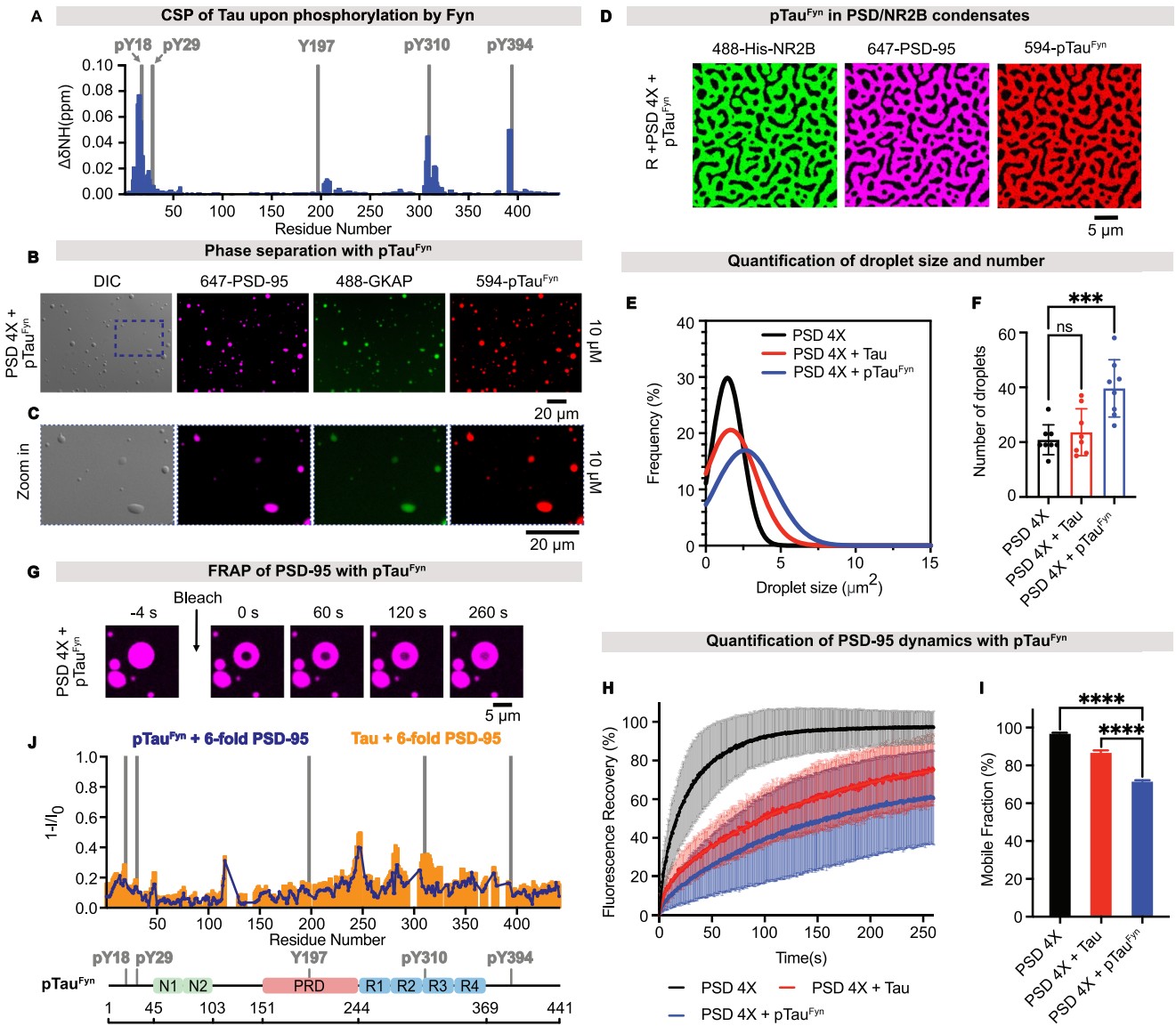

**Fig. 4 | Phosphorylated Tau reduces the dynamic nature of PSD condensates.** **A** NMR chemical shift perturbation analysis of Tau upon Fyn-mediated phosphorylation (blue bars). Gray bars represent the position of Tau's five tyrosine residues. **B, C** DIC and fluorescence microscopy images of phase-separated PSD droplets with Fyn-phosphorylated Tau (pTau$^{Fyn}$). Selected regions are displayed in (**C**). Eleven measurements were done. **D** Confocal fluorescence images of PSD clusters with pTau$^{Fyn}$ on supported membranes. The images were acquired after 15-min incubation at room temperature. Three measurements were done. **E** Size distribution of PSD droplets alone (black), with Tau (red), and with pTau$^{Fyn}$ (blue). **F** Number of droplets formed by the mixture of four PSD scaffold proteins only, with Tau and with pTau$^{Fyn}$. One-way ANOVA analysis was performed (***$p \leq 0.0002$). Same color codes as (**E**). Data are presented as mean values ± SD from $n$ independent measurements. For each condition, $n = 8$. For (**E**) and (**F**), the images corresponding to PSD droplets alone and with Tau are shown in Fig. 1B, C; the images of PSD droplets with pTau$^{Fyn}$ are shown in Fig. 4B, C. **G** Representative FRAP images of PSD-95 in PSD droplets with pTau$^{Fyn}$. PSD-95 was labeled with Alexa 647. **H** FRAP curves in blue represent the average of seven FRAP measurements of PSD-95 dynamics with pTau$^{Fyn}$ from (**G**). The concentration of each protein was 20 μM. FRAP results of PSD-95 without (black) and with Tau (red) are shown for comparison. The corresponding FRAP images of PSD-95 dynamics without and with Tau are shown in Fig. 1D. **I** Mobile fraction of PSD-95 for "PSD 4X + pTau$^{Fyn}$" condition derived from fitted averaged FRAP curve in (**H**), which was averaged from seven FRAP measurements. Error bars represent the standard deviation of curve fits; one-way ANOVA: ****$p \leq 0.0001$. **J** NMR signal intensity profiles of 10 μM pTau$^{Fyn}$ (dark blue lines) and Tau (orange) upon the addition of 6-fold molar excess (60 μM) of PSD-95_SH3GK with I$_0$ being signal intensities of pTau$^{Fyn}$ alone and Tau alone, respectively. Gray bars represent the position of tyrosine residues. The domain organization of pTau$^{Fyn}$ is shown below the intensity profiles. Source data are provided as a source data file.

accumulation of presynaptic Tau, amyloid-beta oligomers, impaired inhibitory interneuron and glial function or genetic risk factors may contribute to synaptic dysfunction in Alzheimer's disease[3]. Notably, however, Tau reduction strongly reduces hyperexcitability in AD mouse lines, induced seizure models, and genetic in vivo models of epilepsy[51], suggesting a critical role of Tau in synaptic dysfunction. Consistent with this hypothesis, clearance of postsynaptic Tau improved cognition in a tauopathy mouse model[32]. Besides lowering

Tau or blocking post-translational modifications of Tau, modalities may thus be developed that block the recruitment of Tau into the postsynaptic density or the multivalent interaction of Tau with PSD-95. Indeed, our proof-of-concept experiments showed that a peptide, which binds with high affinity to the specific Tau-binding site in PSD-95 and inhibits excitatory synaptic activity[43], blocks the Tau-mediated dynamic arrest of in vitro condensates and clusters formed by major PSD scaffold proteins (Fig. 3).

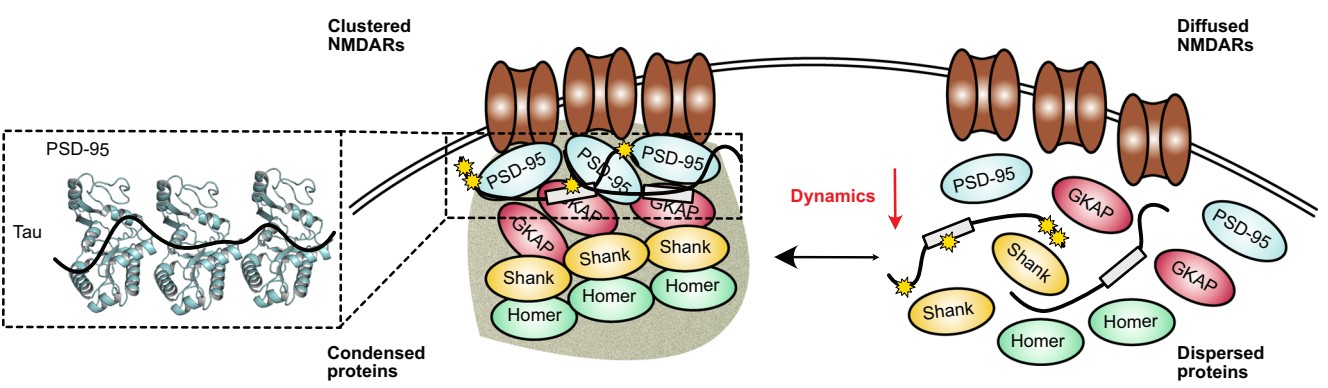

**Fig. 5 | Proposed model illustrating Tau-mediated dynamic arrest of the PSD drives synaptic dysfunction.** At the postsynapse, Tau/pTau^Fyn concentrates in the PSD, binds to PSD-95 through multivalent interactions, and decreases PSD dynamics. Dynamically arrested PSD condensates/clusters enhance the clustering and retention time of NMDA receptors on the membrane.

On the other hand, recent studies have increasingly demonstrated the presence of synaptic Tau in physiological conditions[52] and its various physiological functions within the synapse, including brain-derived neurotrophic factor (BDNF)-induced synaptic plasticity[53], hippocampal neurogenesis[54] and long-term depression[55]. Our study demonstrates that wild-type 2N4R Tau can decrease the PSD dynamics, suggesting a possible physiological role of Tau in stabilizing the major PSD scaffold protein PSD-95 and facilitating the downstream signaling pathways of NMDA receptors.

Compared to wild-type Tau, the recruitment of disease-associated pTau^Fyn increases both the number and size of PSD condensates and also enhances the dynamic arrest of PSD condensates. The size of PSD within a dendritic spine is linearly correlated to the spine size, which is tightly correlated with the synaptic activity strength[56–58]. Whether pTau^Fyn can increase the PSD size within a dendritic spine, hence leading to a hyperactivated synaptic activity in vivo, remains to be elucidated. Therefore, future in vivo experiments and approaches are critical for understanding the molecular process driving PSD formation, characterizing Tau/PSD-95/NMDAR interaction within PSDs, and clarifying the physiological and pathological roles of Tau in PSD regulation.

In conclusion, our investigation reveals that multivalent Tau/PSD-95 interactions modulate the dynamic arrest of in vitro biochemically reconstituted PSD condensates and membrane-bound PSD clusters. Importantly, we have identified an inhibitory peptide to reverse this Tau-mediated dynamic arrest. Together, these findings suggest a key role of Tau in the regulation of PSD dynamics, paving the way for future in vivo investigation and drug development.

## Methods
### Protein expression and purification
The design of the constructs of PSD scaffold proteins and NMDA receptor followed[21]: full-length PSD-95; PSD-95_SH3-GK containing the SH3 and GK domains; GKAP_GBR_PBM containing three GK-binding repeats (GBR) and the PDZ-binding motif (PBM); Shank_-M1718E_PDZ_HBS_CBS_SAM with M1718E mutation and containing the PDZ, Homer-binding sequence (HBS), cortactin-binding sequence (CBS) and sterile alpha motif (SAM); full-length Homer; His$_6$-tagged and untagged GCN4-NR2B.

Genes for the different protein constructs were cloned into modified pET-28a vectors coding for an N-terminal His$_6$-affinity tag followed by a Z2 solubility tag and a TEV cleavage site for tag removal. Recombinant proteins were expressed in Escherichia coli BL21 (DE3)

cells (Novagen). Cells were grown in LB medium supplemented with kanamycin, and protein expression was induced by 0.5 mM IPTG (1 mM for Shank and Homer) when OD600 reached 0.8. After protein expression at 16 °C overnight (or 30 °C for 3 h for PSD-95_SH3-GK; 37 °C for 3 h for His-NR2B; and 25 °C overnight for Shank), cells were harvested by centrifugation with 7000 × $g$ at 4 °C for 30 min. Cell pellets were resuspended in lysis buffer consisting of 50 mM Tris, pH 8.0, 300 mM NaCl, 10 mM imidazole, 2 mM β-mercaptoethanol (6 mM β-mercaptoethanol for PSD-95), 0.5 mM PMSF, 1 mg/mL lysozyme, 5 μg/mL DNase, 1 mM MgCl$_2$, and cOmplete EDTA-free protease inhibitor cocktail. The cells were disrupted by sonication on ice. After sonication, cellular debris were removed by centrifugation at 48,254×$g$ and 4 °C for 40 min. Target proteins present in the supernatant were purified by Ni$^{2+}$ affinity column (GE Healthcare or Qiagen) using 50 mM Tris, pH 8.0, 300 mM NaCl, 10 mM imidazole, and 2 mM β-mercaptoethanol or 6 mM β-mercaptoethanol for PSD-95 as binding buffer and 50 mM Tris, pH 8.0, 300 mM NaCl, 10 mM imidazole, and 2 mM β-mercaptoethanol or 6 mM β-mercaptoethanol for PSD-95 with 300 mM imidazole as elution buffer. Non-specifically bound proteins were removed with an elution buffer. The eluted proteins were dialyzed overnight at 4 °C against TEV cleavage buffer (50 mM Tris, pH 8.0, 150 mM NaCl, 0.1 mM EDTA, 0.5 mM PMSF, 2 mM β-mercaptoethanol and 6 mM β-mercaptoethanol for PSD-95) and cleaved by TEV the next day. After TEV cleavage, non-cleaved proteins and residual His$_6$-tags were removed from the sample using a Ni$^{2+}$ affinity column and collecting the flow through. The target proteins were concentrated and further purified by their molecular weight using size-exclusion chromatography (Superdex 75 or Superdex 200 columns; GE Healthcare) using a buffer containing 50 mM Tris, pH 8.0, 300 mM NaCl, 1 mM EDTA, and 3 mM DTT. For the purification of PSD-95, PSD-95_SH3-GK, Homer and His-NR2B, the proteins were further purified by ion exchange chromatography to remove unwanted contaminants. In addition, His-NR2B purification skipped the TEV cleavage and the second Ni$^{2+}$ affinity chromatography (after TEV cleavage) steps to keep the N-terminal His$_6$-tag.

For the preparation of unlabeled and $^{15}$N-labeled 2N4R human Tau (htau40; UniProt: P10636-8), its gene was inserted into the pNG2 vector (a derivative of pET-3a, Novagen) and expressed in Escherichia coli strain BL21 (DE3). Cells were grown in LB medium supplemented with ampicillin until OD600 reached 0.8. The expression of unlabeled hTau40 was induced with 0.5 mM IPTG at 37 °C for 1 h. For uniformly $^{15}$N labeling of htau40, the protein was expressed once an OD600 of 0.8 was reached, and the cells from 1 L of LB were

transferred to 250 mL of M9 minimal medium supplemented with 2 g $^{15}NH_4Cl$ and 0.5 g ISOGRO-15N as the only nitrogen source and ampicillin. After 1 h of incubation at 37 °C the culture was induced with 1 mM IPTG overnight at 37 °C. Bacterial cells were harvested by centrifugation and the pellets were resuspended in a 20 mM MES buffer at pH 6.8, including 1 mM EGTA, 0.5 mM MgCl₂, 1 mM PMSF, 1 mg/mL lysozyme, 10 µg/mL DNase I, 5 mM DTT and cOmplete EDTA-free protease inhibitor cocktail. Cells were disrupted by French Press on ice. The resulting lysates were supplemented with NaCl to a final concentration of 500 mM and boiled at 98 °C for 20 min. Lysates were cooled down on ice and ultracentrifuged at 127,000 × g and 4 °C for 40 min. DNA of the supernatant was precipitated by adding 20 mg/mL of streptomycin and incubated at 4 °C for 15 min with rotation. After centrifugation at 15,000 × g and 4 °C for 30 min, the pellet was discarded and the resulting supernatant was incubated with 361 mg/mL (NH₄)₂SO₄ at 4 °C for 15 min to precipitate hTau40. Then, precipitated proteins were pelleted by repeating the previous centrifugation step. The pellets containing the protein were resuspended in dialysis buffer (20 mM MES buffer, pH 6.8, 1 mM EDTA, 0.1 mM PMSF, and 2 mM DTT) and dialyzed against the same buffer at 4 °C overnight to remove salts. Filtered dialysate was purified by cation exchange chromatography using a MonoS 10/100 (Cytiva) with a linear gradient from 0% to 60% of elution buffer (binding buffer: 20 mM MES, pH 6.8, 50 mM NaCl, 1 mM EDTA, 2 mM DTT, and 0.1 mM PMSF; elution buffer with the same composition as binding buffer, except with 1 M NaCl). After identifying the fractions containing hTau40 with SDS-PAGE, these were pooled together and concentrated to be further purified by size-exclusion chromatography with PBS supplemented with an additional 500 mM NaCl, 1 mM DTT and 0.1 mM PMSF at pH 7.4 as running buffer. Protein concentrations were determined by Pierce™ BCA protein assay kit (Thermo Scientific) before using them in further experiments.

Protein purity was analyzed by running 4%-20% precast polyacrylamide gels (BIO-RAD) in SDS-containing Tris-Glycine buffer, followed by Coomassie blue staining (Supplementary Fig. 5). Purified proteins were dialyzed against the respective buffers, concentrated, flash frozen and stored at −80 °C until further use.

## Peptide preparation

TMR-NR2B peptides with N-terminal TMR label (amino acid sequence: WARMKQIEDKLEEILSKLYHIENELARIKKLLGSIESDV) were ordered from Peptide Specialty Laboratories GmbH (Heidelberg). TMR-NR2B stocks were prepared by weighting and dissolving the required amount of peptide powders in the phase separation buffer (50 mM Tris, pH 7.8, 100 mM NaCl, 1 mM EDTA, and 5 mM DTT) with 1% DMSO.

pDLS peptide (amino acid sequence: RIRRE(p)SYRRANGQSFDLS) with acetylated N-terminus and amidated C-terminus was synthesized by solid-phase peptide synthesis. The required amount of pDLS powders was dissolved in either phase separation or NMR buffer (50 mM sodium phosphate, pH 6.8, 100 mM Na₂SO₄, and 1 mM DTT).

## Phosphorylation of Tau

For Fyn-mediated phosphorylation of Tau, 100 µM $^{15}N$-labeled Tau was phosphorylated with 2 µM of Fyn kinase (ThermoFisher Scientific) in Fyn phosphorylation buffer (25 mM HEPES, pH 7.4, 5 mM EGTA, 10 mM MgCl₂, 4 mM ATP, and 2 mM DTT). The reaction was incubated with 300 rpm shaking at 30 °C for 24 h. For c-Abl-mediated phosphorylation, 50 µM $^{15}N$-labeled Tau was phosphorylated with 0.15 µM of c-Abl kinase (ThermoFisher Scientific) in c-Abl phosphorylation buffer (25 mM HEPES, pH 7.4, 2 mM EGTA, 5 mM MgCl₂, 2 mM ATP, 0.5 mM PMSF and 2 mM DTT). The reaction was incubated with 300 rpm shaking at 25 °C for 24 h. As Tau is an intrinsically disordered protein, Fyn and c-Abl were inactivated by boiling the reaction mixture at 65 °C for 20 min, followed by high-speed centrifugation at 15,000 × g for

30 min. The supernatant containing pTau$^{Fyn}$ and pTau$^{c-Abl}$ was dialyzed against buffers used in further experiments.

## Phase separation assays for fluid PSD condensates

For phase separation assays, proteins were labeled with Alexa Fluor™ 488, Alexa Fluor™ 594, and Alexa Fluor™ 647 microscale protein labeling kits (ThermoFisher Scientific, Invitrogen). For fluorescent microscopy, small amounts of fluorescently labeled PSD-95, Tau, or pTau$^{Fyn}$ (~0.3 µL) were premixed with unlabeled PSD-95, Tau, and pTau$^{Fyn}$ protein stocks, respectively. Then, PSD scaffold proteins and Tau (or pTau$^{Fyn}$) were mixed at equal molar ratios and diluted in the phase separation buffer (50 mM Tris, pH 7.8, 100 mM NaCl, 1 mM EDTA, and 5 mM DTT) to reach the indicated protein concentrations. No condensate-inducing crowding agent was used in the condensate experiments. For each sample, 5 µL of the reaction was added onto a microscope slide with an 18 mm coverslip. DIC and fluorescent imaging were performed on a Leica DM6B microscope, and a 63x water objective was used to observe and acquire images at room temperature. Micrographs were analyzed using Fiji (Version 2.9.0; NIH).

Droplet sizes and numbers were determined on intensity-normalized images (eight images for each phase separation assay) using the Analyze Particles function in Fiji (NIH). The lower threshold for droplet size was 0.2 µm². Fused droplets and droplets appearing on edges were excluded from the analyses. Droplet size distributions were further analyzed in GraphPad Prism (Version 9.3.1). to plot frequency distribution histograms, which were fitted with Gaussian distribution curves. One-way ANOVA was performed using GraphPad Prism (***$p \leq 0.0002$).

## Fluorescence recovery after photobleaching (FRAP)

The dynamic nature of phase-separated PSD and PSD/NR2B droplets was analyzed using FRAP. Phase separation of PSD proteins was induced as described above. PSD/NR2B droplets were formed by mixing 1 µM TMR-labeled NR2B peptide with PSD scaffold proteins and different concentrations of Tau.

FRAP was recorded on a Leica TCS SP8 confocal microscope using a 63x oil objective at room temperature. The circular region of interest (ROI) with a diameter of 1.9 µm was selected in the middle of each droplet. After recording 3 frames (with a frame rate of 523 ms), the fluorescence of Alexa 647-labeled PSD-95 and TMR-labeled NR2B in ROI was photobleached by a 488-argon and a DPSS 561 laser beam at 50% laser power for 5 frames, respectively. After photobleaching, 500 frames were recorded to capture the fluorescence recovery. The images were analyzed using Fiji (NIH). Each FRAP curve was calculated according to:

$$FRAP = \frac{I_{Postbleach} - I_{Background}}{I_{Av.Prebleach}} \quad (1)$$

where $I_{Postbleach}$ indicates the measured fluorescence intensity of ROI after bleaching, $I_{Av.Prebleach}$ indicates the average intensity of ROI before bleaching. Both intensities were corrected by background subtraction.

The calculated results were further corrected by multiplying the acquisition bleaching correction factor (ABCF), which was calculated according to:

$$ABCF = \frac{I_{Av.(Reference.prebleach)}}{I_{Reference} - I_{Background}} \quad (2)$$

where $I_{Reference}$ indicates the fluorescence intensity of the whole droplet area, $I_{Av.(Reference.prebleach)}$ indicates the averaged $I_{Reference}$ before bleaching. The values were corrected by background subtraction. Intensity differences between frames before ($I_{Prebleach}$) and just after bleaching ($I_{Postbleach}$ at time 0; called as $I_i$) were normalized to 100%

according to:

$$\text{Normalization} = \frac{I_{(t)} - \text{min.Intensity Value}}{1 - \text{min.Intensity Value}} \quad (3)$$

Subsequently, FRAP curves were averaged and fitted with a bi-exponential function using GraphPad Prism:

$$I_{fit} = I_i + a_{fast} * (1 - \exp(-K_{fast} * t)) + a_{slow} * (1 - \exp(-K_{slow} * t)) \quad (4)$$

$$a_{fast} = (I_{Plateau} - I_i) * \text{PercentFast} * 0.01 \quad (5)$$

$$a_{slow} = (I_{Plateau} - I_i) * (100 - \text{PercentFast}) * 0.01 \quad (6)$$

where $I_i$ were constrained to a constant value of 0. $I_{plateau}$ indicates the fluorescence intensity at infinite times and was constrained to be less than 100. $K_{fast}$ and $K_{slow}$ indicate the rate constants of the faster and slower components, respectively.

Mobile fractions were derived from curve fits based on the following equation:

$$\text{Mobile fraction} = \frac{I_{plateau} - I_i}{I_{prebleach} - I_i} \quad (7)$$

Welch's $t$-test was performed to compare the mobile fractions of PSD 4X condensates without and with Tau (****$p \leq 0.0001$). For other experiments (as indicated), one-way ANOVA analyses were performed (**$p \leq 0.0021$, ***$p \leq 0.0002$, ****$p \leq 0.0001$).

## Membrane-anchored PSD clustering

Phospholipids, including POPC, 4% Ni²⁺-NTA-DGS, 0.1% PEG5000PE, and 0.1% rhodamine-PE, were mixed in chloroform in glass bottles and dried under vacuum for 2 h. 500 μL to 1 mL buffer (50 mM Tris-HCl, pH 7.8, 100 mM NaCl) was added to the dried lipid film to reach a final lipid concentration of 1 mM and resuspended with 180 rpm shaking at 37 °C for 1 h. The lipid mixture was then transferred to an Eppendorf tube and went through freeze-thaw for 15 runs until the lipid mixture became clear to form small unilamellar vesicles (SUVs). The SUVs were further clarified by centrifugation at 17,000 × $g$ for 30 min and stored at 4 °C for 1 week.

For the preparation of supported lipid bilayers, glass-bottomed 96-well plates (Greiner, Product No. 655891) were pre-cleaned with 2% Hellmanex II overnight and 6 M NaOH for 30 min twice at room temperature. Before adding SUVs, wells were equilibrated with buffer (50 mM Tris-HCl, pH 7.8, 100 mM NaCl) for 5 min, and left 60 μL buffer in the well. Next, 20 μL 1 mM SUVs was added and incubated for 20 min. Following this, 20 μL 5 M NaCl was added for another 20 min for the SUVs to further collapse onto the glass bottom. Excess SUVs were washed away by pipetting in and out buffer (50 mM Tris-HCl, pH 7.8, 100 mM NaCl) eight times. The quality of supported lipid bilayers was controlled by FRAP under the confocal microscope.

Supported lipid bilayers with a certain amount of Ni²⁺-NTA-DGS lipid were blocked with 1 mg/mL BSA for 30 min. Then, 500 nM NR2B receptors were added and incubated for 30 min. Excessive receptors were washed away by pipetting in and out buffer (50 mM Tris-HCl, pH 7.8, 100 mM NaCl, 1 mM TCEP) eight times. Next, 1 μM of each PSD 4X scaffold protein was added with and without a certain amount of Tau/pTau^Fyn for 15 min to form membrane-associated PSD clusters. Three images for each well were taken randomly under the confocal microscope. Time lapses were taken with a frame rate of 30 s for 15 min under the confocal microscope (Abberior Instruments, Göttingen, Germany).

For the partition assay of Tau and pTau^Fyn, 1 μM of each PSD 4X scaffold protein with and without Tau or pTau^Fyn were added to supported lipid bilayers functionalized with NR2B receptors for 15 min. Three images for each well were taken randomly under the confocal microscope.

FRAP and confocal imaging on supported membranes was performed on a commercial confocal STED microscope (Abberior Instruments, Göttingen, Germany) with pulsed laser excitation (490 nm, 560 nm, 640 nm, 40 MHz) and 60x water or 100x oil objectives (Olympus). For FRAP, regions of interest were bleached using a 405 nm diode with 1.5 mW with 100 μs pixel dwell time. Pre-bleach and post-bleach images were acquired with 490, 560, and 640 nm laser with a frame rate of 2 s for 5 min. Recovery data were normalized to a reference ROI outside the bleached area.

FRAP traces were evaluated and fitted in GraphPad Prism. Calculations for curve fits, mobile fractions, and one-way ANOVA tests of PSD-95 dynamics on the membrane were performed in the same manner as for the FRAP experiments for condensates in solution. However, for the His-NR2B dynamics (Supplementary Fig. 2), the FRAP curves were fitted to a mono-exponential equation:

$$I_{fit} = I_i + (I_{plateau} - I_i) * (1 - \exp(-K * t)) \quad (8)$$

Mobile fractions were derived from the fitted curves based on Eq. 7. Welch's $t$-tests were performed to compare the mobile fractions of FRAP experiments with 488-His-NR2B on membranes.

## NMR spectroscopy

Two-dimensional ¹H-¹⁵N HSQC experiments were recorded at 278 K on an 800 MHz spectrometer (Bruker) in NMR buffer (50 mM sodium phosphate, pH 6.8, 100 mM Na₂SO₄, and 1 mM DTT) supplemented with 10% D₂O. The same experimental parameters were used throughout the study except for the number of scans: each spectrum was recorded with 48 scans for the experiments involving 20 μM ¹⁵N-labeled Tau and with 192 scans for the experiments involving 10 μM ¹⁵N-labeled Tau/¹⁵N-labeled pTau^Fyn/¹⁵N-labeled pTau^c-Abl/¹⁵N-labeled Tau_Y394N. For all experiments, we set 2048 and 256 increments in the ¹H and ¹⁵N dimensions, respectively. Spectra were processed with TopSpin 4.0.6 (Bruker) and analyzed using Sparky (NMRFAM-Sparky 1.470 powered by Sparky 3.190; ref. [59]). Assignments of ¹H-¹⁵N cross-peaks of Tau were previously determined[60].

To investigate the multivalent interaction between PSD-95 and Tau (or pTau^Fyn) and how pDLS affect this interaction, we analyzed the peak intensity changes according to:

$$\text{Peak intensity changes} = 1 - \frac{I}{I_0} \quad (9)$$

where I represents the NMR peak intensity of Tau or pTau^Fyn in the presence of other proteins (i.e., PSD-95, PSD-95_SH3-GK, and pDLS); $I_0$ represents the peak intensity of Tau/ pTau^Fyn alone. For NMR intensity analyses of HSQC spectra of ¹⁵N-labeled Tau/pTau^Fyn with PSD-95 or PSD-95_SH3GK, independent samples were prepared for each titration point and the highest intensity ratio ($I/I_0$) was normalized to 1. For HSQC titration experiments without or with the pDLS peptide, increasing concentrations of pDLS were titrated into the same NMR sample containing 20 μM ¹⁵N-labeled Tau with 120 μM PSD-95_SH3-GK, and intensity plots were normalized according to the averaged peak intensity of tau residues D38 to G186.

To identify tyrosine residues of Tau which are phosphorylated by Fyn, ¹H-¹⁵N HSQC spectra of 20 μM ¹⁵N-labeled Tau and 20 μM ¹⁵N-labeled pTau^Fyn were recorded. As for the tyrosine residues of Tau phosphorylated by c-Abl, due to limited protein stock, ¹H-¹⁵N HSQC spectra of 10 μM ¹⁵N-labeled Tau and 10 μM ¹⁵N-labeled pTau^c-Abl were recorded. The experimental conditions and parameters were the same as described above. For pTau^Fyn and pTau^c-Abl cross-peaks that are closest to the position of unmodified Tau were selected. Chemical

shift perturbation (CSP) was calculated according to:

$$CSP = \sqrt{\frac{\delta H^2 + \left(\frac{\delta N}{5}\right)^2}{2}} \qquad (10)$$

where $\delta H$ and $\delta N$ represent the chemical shift changes of proton and nitrogen nuclei, respectively. CSP and intensity plots were smoothed using a sliding average of three neighboring residues.

## AlphaFold

The three-dimensional protein structures of PSD-95_SH3-GK with Tau peptides were predicted using AlphaFold2[42] as implemented on the Google ColabFold notebook[61] available at https://colab.research.google.com/github/sokrypton/ColabFold/blob/main/AlphaFold2.ipynb. Amber-relaxed, top-ranked AlphaFold2 structures were superposed with the X-ray crystal structure of PSD-95_SH3-GK/GKI-QSF (PDB code: 5ypr) using Pymol (The PyMOL Molecular Graphics System, Version 2.0 Schrödinger, LLC).

## Statistics and reproducibility

All relevant information regarding statistical analysis, including the test statistic, confidence intervals, degrees of freedom, effect sizes and exact *p*-values, is provided in the source data file. Similarly, all relevant information regarding curve fitting, including the plateau and standard error, is provided in the source data file. No statistical method was used to predetermine the sample size. No data were excluded from the analyses. The experiments were not randomized. The investigators were not blinded to allocation during experiments and outcome assessment.

## Reporting summary

Further information on research design is available in the Nature Portfolio Reporting Summary linked to this article.

## Data availability

All data needed to evaluate the conclusions in the paper are present in the paper and/or the Supplementary Materials. The cited PDB id 5ypr[43] is publicly available in the PDB. Source data are provided with this paper.

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

## Acknowledgements

We thank Kerstin Overkamp for peptide synthesis, Ines Boussena for help with protein production, and the light microscopy facility of the Max Planck Institute for Multidisciplinary Sciences for microscope access. M.Z. was supported by the European Research Council (ERC) under the EU Horizon 2020 research and innovation programme (grant agreement No. 787679). A.H. was supported by Deutsche Forschungsgemeinschaft (DFG) project number 402723784-SPP2191.

## Author contributions

Z.S. produced protein and performed biochemical/biophysical experiments, phase separation assays, and NMR spectroscopy. D.S. performed membrane-anchored PSD/NR2B clustering assays. A.S and M.Z. supervised biochemical/biophysical experiments, phase separation assays, and NMR spectroscopy. S.J.V. produced protein and performed FRAP experiments. M.-S.C-O. prepared recombinant protein. S.B. performed biophysical experiments. A.H. supervised membrane-anchored PSD/NR2B clustering assays. All authors contributed to manuscript preparation. M.Z. designed the project.

## Funding

## Competing interests

One of the authors, A.S., is an editor on the staff of Nature Communications, but was not in any way involved in the journal review process. All the other authors declare no competing interests.
