## [Peer Review File · Nature Communications]

Multivalent Tau/PSD-95 interactions arrest in vitro condensates and clusters mimicking the postsynaptic densityReviewer #1 (Remarks to the Author):

This is a very short paper from a leading structural biology lab investigating the potential interactions of tau or phospho-tau with PSD95 and the cytoplasmic sequence of NR2B. The data convincingly demonstrated an interaction of these proteins in condensates that presumably represent phase-separated aggregates, and show that likely the GK domain of PSD95 is involved. However, there is no biology here. Similar to much of the phase separation literature, we don't actually know if any of this happens in vivo. To this date, it has not been conclusively shown that PSD95 interacts with NMDA receptors in a cell. There is still doubt whether postsynaptic phase separation is a physiological event. Given the high concentrations of proteins involved in these in vitro interactions and the enduring questions about the functions of tau and of PSD95 I feel that this paper is more appropriate for a journal like 'Structure' or 'Biophysical Journal'. What is the minimum that would need to be done to at least suggest that the current findings are biologically relevant? It should be demonstrated by super-resolution microscopy that tau is actually co-localized with PSD95 and NMDA receptors. Genetic manipulations should demonstrate that their localizations depend on each other. Changing the phosphorylation of tau should alter these interactions. Ideally an approach should be identified that allows testing whether the observed condensates are occurring in a cell or are simply an artifact of high protein concentrations in a test tube.

Reviewer #2 (Remarks to the Author):

Evidence is accumulating that tau has potentially many physiological and pathological roles in the brain and dysregulation of synaptic functionality is of high interest to the field. The work by Shen et al presents several interesting findings through a series of in vitro phase separation assays with 4-5 postsynaptic density (PSD)-associated proteins. The studies provide some novel insights into potential effects of tau and fyn phosphorylated tau on the dynamics of PSD95 and NMDA receptors in such a setting. The work is generally well done and rigorous. Better understanding the interactions and mechanisms of potential tau-mediated synaptic regulation/dysregulation is a critical need.

However, several significant issues must be addressed with the manuscript in its current form. Among them, three are considered particularly major. First, there are several misstatements in the paper, specifically in regards to normal tau distribution and the unproven notion of tau missorting in disease. Several comments (highlighted below) align with the dogmatic falsehood that tau is not found at normal levels in the somatodendritic compartment in the adult brain. There are several overstatements where such concepts are presented as though they are established fact in human disease. Two salient points to this major concern are 1) the impact of work presented here does not rely on or need there to be no/low tau in the somatodendritic compartment normally and does not require tau missorting to be true, and 2) these issues can be easily resolved. The second major concern is somewhat aligned. Throughout the authors stretch the interpretation of their data/work and that of others too far. In addition, the data and conclusions need to be interpreted within the bounds of what has been done. These are all in vitro, artificial experimental setups and they are not functional postsynaptic densities. Specific examples of these issues are highlighted below. Third, the lack of in-cell, specifically neurons, evidence of tau-mediated regulation or dysregulation of PSD-related proteins in condensate structures is a major concern and significantly dampens the impact of the work. Without such experiments the significance and physiological relevance of the findings remain uncertain limiting the impact of the work.

The title needs to reflect key aspects of the work more accurately. The work does not look at postsynaptic densities – when this term is used alone this implies that actual in-neuron PSDs. In addition, the title should indicate that this work is entirely in vitro (i.e. not in-cells). Finally, the intended meaning with the word “over” is not clear.

The authors repeatedly put forth the idea of tau “missorting” as though it is established fact and an essential component of the human disease process. Indeed, this idea and the papers provided as references are largely based on the false assumption that tau is not in the somatodendritic

compartment of neurons in the adult brain. If the authors are reporting results from papers that presume to assess tau missorting they need to be more explicit about exactly what was done (e.g. in primary cultured neurons, with tau overexpression, etc.). It is noteworthy that none of these findings are from human disease studies and most of this type of work uses developing cultured neurons. Accordingly, the following statements need to be removed or revised significantly:

- "In Alzheimer's disease, hyperphosphorylated Tau is missorted into dendritic spines, correlating with synaptic dysfunction and cognitive impairment." "Key to Tau-mediated synaptic dysfunction is the redistribution of Tau from axons to the somatodendritic compartment and in particular to the postsynapse." "Notably, PSD condensate recruitment and dynamic arrest occurred with monomeric Tau, suggesting that Tau missorting to the postsynapse is responsible for early functional deficits prior to Tau oligomerization and aggregation." None of these are well-supported statements.
- "Tau phosphorylated at Y18 by the tyrosine kinase Fyn, specifically triggers Tau detachment from microtubules and Tau missorting to the postsynapse." The references included do not show that there is an effect on microtubule binding affinity or redistribution of Fyn phosphorylated tau proteins, perhaps the authors can clarify exactly what findings show this very specific chain of events.
- "We therefore investigate in the current work whether missorted Tau concentrates in PSD condensates, changes PSD material properties and affects NMDA receptor dynamics." There are no measures of tau missorting in this paper. What exactly is "missorted tau"?
- "Our findings provide a neurotoxic mechanism for prolonged clustering and activity of NMDA receptors on the postsynaptic membrane and thus for excitotoxicity-associated cognitive dysfunction in Alzheimer's disease." Neurotoxicity and receptor activity are not measured in this manuscript, nor is there any use of Alzheimer's disease samples.
- "To investigate the activity of Tau at the postsynapse,..." Again, the lack of accurately describing what is being measured must be modified. There are no assays looking at tau in the postsynapse.
- "Physiological Tau locates to axons where it associates with microtubules." This statement implies tau is an axon-specific protein and must be changed to reflect that tau is located throughout the neuron where it binds microtubules.
- "In Alzheimer's disease, phosphorylation of Tau by the tyrosine kinase Fyn dissociates Tau from the microtubules resulting in Tau missorting to the postsynapse (Fig. 4A)." This statement is false, this has not been shown in Alzheimer's disease.
- "The data suggest that phosphorylation and other post-translational modifications may predominantly promote detachment of Tau from microtubules and subsequent redistribution of Tau from axons to dendritic spines." The data in this manuscript do not speak to these points at all. Also, tau does not detach from microtubules, some specific modifications can reduce binding affinity. The unsubstantiated dogma of tau redistribution is noted above.
- The statement that in vitro-formed 4-5 protein condensates are "representing the postsynaptic density (PSD)" is an overstretching assertion.

The studies reported do not clearly rule out the possibility that tau is impacting one of the other in vitro PSD condensate proteins as the authors have only assessed the impact of tau on PSD95 and N2RB. Without looking at each of the other proteins this remains unclear.

The pDLS peptide competition studies with NMR and droplet assays are a good initial set of studies showing some degree of domain specificity. However, a set of experiments that more directly demonstrate the specificity of the condensate interaction between tau and PSD95 (and potentially other constituents, per comment above) would strengthen the work. For example, testing deletion or point mutation versions of tau, PSD95, N2RB and other PSD proteins (potentially) that are predicted to interfere with their interaction in condensates would strengthen the work. One would predict that ideally they can phase separate on their own but would not coalesce in the same condensate.

The authors state that tau had a dose-dependent effect on mobile fractions in Figures 1 and 2, but the data do not support a dose-dependent effect. To make this conclusion, each dose must be significantly different from the others – this does not appear to be the case based on statistics shown.

The authors did not mention whether Y197 was modified in the pTau-Fyn sample. Moreover, if Fyn-mediated phosphorylation is a critical component to tau's role in the PSD then the work would

be strengthened by determining the specificity of the effects for each of the 5 available tyrosine residues in tau.

Based upon the methods section, it does not appear that a condensate-inducing crowding agent (e.g. PEG, RNA, etc.) was used in the fluid condensate experiments. Please clarify.

It's striking that all the data are interpreted in the context of pathological tau or toxicity mechanisms. Given the work that was done, it is unclear how the authors have ruled out a that these tau-mediated effects on PSD condensates reflect a potential normal physiological function.

There doesn't appear to be any validation of recombinant proteins.

Figure 1 legend: "The Alzheimer's disease-associated protein Tau...". This is an inappropriate way to talk about tau protein.

Figure 1C – it is not clear how it was determined that the arrows are indicating/depicting fusing droplets.

As noted above, the suggested distribution of tau in Figure 4A is inaccurate and the notion of tau mislocalization being responsible for somatodendritic tau (in disease) is not supported by evidence in human disease and is based largely on the assumption it is not normally in the somatodendritic compartment and data from cultured neurons.

Figure 4B – it is not Y187 it is Y197.

Figure 5. The model stretches the data in this work too far. The study was focused on tau altering the dynamics of a small set of PSD-associated proteins using in vitro phase separation experiments. This does not inform the role of microtubule binding, effects of fyn phosphorylation on microtubule binding, and synaptic functional changes.

Reviewer #3 (Remarks to the Author):

The manuscript by Shen et al. describes the interaction of tau with Post-synaptic density proteins in vitro and the formation of condensates. In the presence of tau, the recovery after photobleaching is reduced and this is even more so in the presence of Fyn phosphorylated tau. The authors nicely demonstrate that the reduced recovery is partially rescued in the presence of a design peptide which competes for the supposed binding site of PSD95. The experimental work is then very nicely supported by NMR studies to investigate interactions between the molecular components. Overall, this is an interesting piece of work that provides insight into the mechanisms by which tau may influence the properties of the PSD resulting in alteration in NMDA receptor activity. The paper ends with a suggested mechanism that relates the increase in phosphorylation of tau with its reduced binding to microtubules and mislocalisation to the PSD.

Importantly, I believe the authors may want to consider the different isoforms of tau that are present in the cell and it is not possible to assume that all are the same. It may be that the tau that associates with microtubules differs from the one that associates with PSD and there is nothing in this paper that can provide further insights into this. Therefore, some of the "definite" language regarding temporal processes (e.g when something is phosphorylated and where it goes) is speculative. This does not diminish the impressive work provided here but merely points to the care needed when referred to "tau" as a general term.

Also, the work provided here is in vitro and is extremely nicely performed at relatively low concentrations. However, it is not possible to definitively say this is what is happening in the context of a cellular milieu so again, I would advise some rephrasing to account for this.

I have some Comments which really refer to the early part of the paper:

1. The authors mention in the introduction that phosphorylation occurs prior to aggregation. However, it is far from determined exactly when phosphorylation occurs on tau and although they refer to three papers, they could easily find more that state the opposite view. It is not clear which

tau isoform they are referring to and it is not certain whether all tau aggregates, and what the temporal organisation of the process may be. It may be that the tau that associates in PSD never forms aggregates.

2. Results: The type of tau being used should be stated here when it is first introduced (without requiring reference to the methods)

3. Figure 1B shows droplets for the mixture of the PSD proteins and then the addition of tau also shows droplets. It is not entirely clear to me what the authors mean when they say the four components of PSD form LLPS, and then the tau leads to spherical droplets. What is the difference. The authors say that larger droplets are formed in the presence of tau (1C) but there is no quantification and a single image is provided. Either this observation should be removed, or justified with quantification.

4.1H shows curves of FRAP and it is interesting to note very little difference between 0 and 0.5 tau. The curves overlap and even cross. In 1G, there appears to be a "significant" difference. In the legend, it states that 4-6 curves are included for FRAP but the number of images used for 1G is not included. Why would these show different differences?

5. As a non-NMR specialist this could be a naive question. Can the authors see all 441 residues in the NMR spectra? I ask this because I know that purification of 2N4R is notoriously difficult and gives rise to truncations and fragmentation of the protein. It might have been nice to show some data on the purity of the tau

6. In the discussion the authors state "The data suggest that phosphorylation and other post-translational modifications may predominantly promote detachment of Tau from microtubules and subsequent redistribution of Tau from axons to dendritic spines 11,12". Here they do refer to literature but the sentence suggests they have data that relates to this here. They don't and this sentence requires rewording so it is clear this is related to literature findings not those included in this paper.

MINOR

1. Abstract: I think arrest should have an s on the end (depending on whether TAU is a plural or singular term).

Reviewer #4 (Remarks to the Author):

This article investigates a possible mechanism by which the protein Tau can cause synaptic dysfunction in Alzheimer's disease (AD) prior to neuronal death. The premise of this work is that Tau, which is usually associated with microtubuli, gets missorted in AD to the postsynapse. Here, the authors determine possible effects of Tau on protein condensates, particularly on PSD condensates, which are known to form at the postsynapse and organize the location and dynamics of the NMDA receptor. They show that Tau co-condensates with the PSD proteins and that the presence of Tau decreases the mobility of these condensates. Phosphorylated Tau increases this effect. They further determine the binding interface of Tau with PSD-95 and show that a PSD-95 peptide can competitively reverse the effects of Tau on the condensates.

This thoroughly conducted study tackles the important and timely question if proteins involved in AD can have an effect on neuronal protein condensate formation. I recommend its publication after a few concerns have been addressed.

Major comments

My major concern is that the manuscript does not show a direct effect of Tau on NMDA receptor mobility (Fig. S2). This is by itself not a problem but the discussion section later suggests that the paper did show "Tau-induced decrease in NMDA receptor mobility" because "reduced PSD-95 dynamics results in decreased NMDA receptor mobility." (Page 12). This is not necessarily the

case, in fact the authors show that PSD-95 has a much higher mobility in the NR2B/PSD condensates than NR2B. Therefore, the data rather suggest that Tau affects the PSD portion of the NR2B/PSD condensates but not the NR2B construct.

Minor comments

Page 2: "an electron-dense molecular cluster" does this refer to EM images? Unclear in context.

Page 6: "2N4R Tau" which Tau constructs were used in the previous experiments?

Fig. 3C discussed on page 7: Because I0 is buried under the X-bar and not visible at all, it is a bit strange to give it its own legend (same with 3F). The text on page 7 talks interchangeably of line intensity (which is measured) and line broadening (which is not measured).

Page 9: "does not require binding of Fyn to Tau" unclear.

Page 11: "The data suggest that phosphorylation and other post-translational modifications may predominantly promote detachment of Tau from microtubules and subsequent redistribution of Tau from axons to dendritic spines" What data are referred to? The data in the paper do not show any of this.

We would like to express our gratitude to the four referees for their insightful evaluation of our manuscript and for providing us with many helpful comments and suggestions to improve it. We carefully considered the referees' comments and have made the necessary modifications accordingly. We believe the suggested improvements have significantly strengthened the overall quality of the work.

Reviewer #1:

This is a very short paper from a leading structural biology lab investigating the potential interactions of tau or phospho-tau with PSD95 and the cytoplasmic sequence of NR2B. The data convincingly demonstrated an interaction of these proteins in condensates that presumably represent phase-separated aggregates, and show that likely the GK domain of PSD95 is involved. However, there is no biology here. Similar to much of the phase separation literature, we don't actually know if any of this happens in vivo.

Reply: We would like to thank the referee for the careful evaluation and for encouraging us to improve our manuscript. We comprehend the referee's concern regarding the biological relevance of our work. Below, we have provided our response to the referee's comments (written in blue).

To this date, it has not been conclusively shown that PSD95 interacts with NMDA receptors in a cell.

Reply: We thank the reviewer for the comment. We added the citation (Kornau et al., 1995) regarding PSD-95/NMDAR interaction in the revised version of the manuscript, indicating that:

- the C-terminal region of NMDA receptor subunit 2B (NR2B) and PSD-95 co-immunoprecipitate from HEK 293 cells
- NR2B and PSD-95 co-localize in culture hippocampal neurons
- *In situ* hybridization showed that NR2B and PSD-95 co-localize in rat brain

There is still doubt whether postsynaptic phase separation is a physiological event.

Reply: We agree with the reviewer that further experimental data are required to investigate the material properties of the post-synaptic density and whether phase separation is at all or to which degree contributing to its formation in vivo. This is, however, beyond the scope of the current manuscript.

Because we use different assays in our work (phase separation in solution as well as "clustering" on membranes), we are convinced that even if phase separation would not be critical for the formation and/or stability of the postsynaptic density in vivo, but instead some other mechanism of protein clustering is relevant, the data reported in the current manuscript (such as the influence of Tau on the clustering of PSD scaffold proteins on the membrane) provides an important advance in our mechanistic understanding of the role of Tau at the postsynapse. To clarify that our data obtained with membranes (e.g. Fig. 2), report primarily on protein clustering, we replaced everywhere in the manuscript "*membrane-associated condensates*" by "*membrane-associated clusters*". Indeed, in the methods section and author contributions we previously already used the term "*membrane-anchored PSD/NR2B clustering assays*." In addition, we explicitly refer in the title and abstract of the revised manuscript to "*condensates/clusters*".

Given the high concentrations of proteins involved in these in vitro interactions and the enduring questions about the functions of tau and of PSD95 I feel that this paper is more appropriate for a journal like 'Structure' or 'Biophysical Journal'. What is the minimum that would need to be done to at least suggest that the current findings are biologically relevant? It should be demonstrated by super-resolution microscopy that tau is actually co-localized with PSD95 and NMDA receptors. Genetic manipulations should demonstrate that their localizations depend on each other. Changing the phosphorylation of tau should alter these interactions. Ideally an approach should be identified

that allows testing whether the observed condensates are occurring in a cell or are simply an artifact of high protein concentrations in a test tube.

Reply: We thank the reviewer for the comment. Some of the questions raised by the reviewer were addressed in a recent publication by Colom-Cadena et al., 2023, which is now cited in the revised version of the manuscript:

- Colom-Cadena et al., investigated the accumulated Tau at the postsynaptic site by using human postmortem brain samples and performing array tomography
- All of the Tau species tested (oligomeric, misfolded and phosphorylated) colocalized with PSD-95 at the postsynaptic site.
- The co-localization of Tau/PSD-95 was further confirmed by super-resolution microscopy techniques, including dSTORM and DNA-PAINT.

Consistent with these findings, Lilek et al. demonstrated by immunofluorescence that PSD-95 and pTau214 colocalization increases during AD progression across clinically stratified groups (normal, aMCI, AD) in frontal cortex. Additionally, Dejanovic et al. found elevated signals of AT8, AT100, and AT180 antibodies (which recognize phosphoepitopes that are abundant in pathological aggregated and filamentous Tau of human AD) in Tau transgenic mice, which were markedly enriched in PSD fractions. These studies are cited in the revised version of the manuscript and their findings are stated in the introduction:

“Tau is located throughout the neuron and regulates the dynamics and organization of microtubules. Additionally, Tau can be found at the postsynapse where it might interact with PSD-95. Different Tau species (oligomeric, misfolded and phosphorylated) colocalize with PSD-95 at the postsynaptic site in human postmortem samples. Consistent with these findings, colocalization of phosphorylated Tau and PSD-95 increases during Alzheimer’s disease progression across clinically stratified groups (normal, mild cognitive impairment, Alzheimer’s disease) in frontal cortex. Additionally, signals of AT8, AT100, and AT180 antibodies, which recognize phosphoepitopes that are abundant in pathological aggregated and filamentous Tau of Alzheimer’s disease, are elevated in Tau transgenic mice, in particular in PSD fractions.”

Reviewer #2:

Evidence is accumulating that tau has potentially many physiological and pathological roles in the brain and dysregulation of synaptic functionality is of high interest to the field. The work by Shen et al presents several interesting findings through a series of in vitro phase separation assays with 4-5 postsynaptic density (PSD)-associated proteins. The studies provide some novel insights into potential effects of tau and fyn phosphorylated tau on the dynamics of PSD95 and NMDA receptors in such a setting. The work is generally well done and rigorous. Better understanding the interactions and mechanisms of potential tau-mediated synaptic regulation/dysregulation is a critical need.

Reply: We thank the reviewer for the positive assessment of our work and for the helpful suggestions for improvements.

However, several significant issues must be addressed with the manuscript in its current form. Among them, three are considered particularly major. First, there are several misstatements in the paper, specifically in regards to normal tau distribution and the unproven notion of tau missorting in disease. Several comments (highlighted below) align with the dogmatic falsehood that tau is not found at normal levels in the somatodendritic compartment in the adult brain. There are several overstatements where such concepts are presented as though they are established fact in human disease. Two salient points to this major concern are 1) the impact of work presented here does not rely on or need there to be no/low tau in the somatodendritic compartment normally and does not require tau missorting to be true, and 2) these issues can be easily resolved. The second major

concern is somewhat aligned. Throughout the authors stretch the interpretation of their data/work and that of others too far. In addition, the data and conclusions need to be interpreted within the bounds of what has been done. These are all in vitro, artificial experimental setups and they are not functional postsynaptic densities. Specific examples of these issues are highlighted below. Third, the lack of in-cell, specifically neurons, evidence of tau-mediated regulation or dysregulation of PSD-related proteins in condensate structures is a major concern and significantly dampens the impact of the work. Without such experiments the significance and physiological relevance of the findings remain uncertain limiting the impact of the work.

Reply: We thank the referee for the constructive feedback. As shown below, we have modified and improved our manuscript based on these comments.

The title needs to reflect key aspects of the work more accurately. The work does not look at postsynaptic densities – when this term is used alone this implies that actual in-neuron PSDs. In addition, the title should indicate that this work is entirely in vitro (i.e. not in-cells). Finally, the intended meaning with the word “over” is not clear.

Reply: Thanks for the suggestion. We modified the title accordingly: *“Multivalent Tau/PSD-95 interactions arrest in vitro condensates and clusters mimicking the postsynaptic density”*.

The authors repeatedly put forth the idea of tau “missorting” as though it is established fact and an essential component of the human disease process. Indeed, this idea and the papers provided as references are largely based on the false assumption that tau is not in the somatodendritic compartment of neurons in the adult brain. If the authors are reporting results from papers that presume to assess tau missorting they need to be more explicit about exactly what was done (e.g. in primary cultured neurons, with tau overexpression, etc.). It is noteworthy that none of these findings are from human disease studies and most of this type of work uses developing cultured neurons. Accordingly, the following statements need to be removed or revised significantly:

- “In Alzheimer’s disease, hyperphosphorylated Tau is missorted into dendritic spines, correlating with synaptic dysfunction and cognitive impairment.” “Key to Tau-mediated synaptic dysfunction is the redistribution of Tau from axons to the somatodendritic compartment and in particular to the postsynapse.” “Notably, PSD condensate recruitment and dynamic arrest occurred with monomeric Tau, suggesting that Tau missorting to the postsynapse is responsible for early functional deficits prior to Tau oligomerization and aggregation.” None of these are well-supported statements.

Reply: We understand the reviewer’s concern and removed these statements according to the suggestions.

“Tau phosphorylated at Y18 by the tyrosine kinase Fyn, specifically triggers Tau detachment from microtubules and Tau missorting to the postsynapse.” The references included do not show that there is an effect on microtubule binding affinity or redistribution of Fyn phosphorylated tau proteins, perhaps the authors can clarify exactly what findings show this very specific chain of events.

Reply: We thank the referee for the suggestion. We removed this sentence.

“We therefore investigate in the current work whether missorted Tau concentrates in PSD condensates, changes PSD material properties and affects NMDA receptor dynamics.” There are no measures of tau missorting in this paper. What exactly is “missorted tau”?

Reply: We thank the referee for pointing this out. We removed “missorted/missorting” fully from the manuscript.

“Our findings provide a neurotoxic mechanism for prolonged clustering and activity of NMDA receptors on the postsynaptic membrane and thus for excitotoxicity-associated cognitive dysfunction in Alzheimer’s disease.” Neurotoxicity and receptor activity are not measured in this manuscript, nor is there any use of Alzheimer’s disease samples.

Reply: Thanks for the suggestion. We removed this sentence and any associated schematic representations.

“To investigate the activity of Tau at the postsynapse,...” Again, the lack of accurately describing what is being measured must be modified. There are no assays looking at tau in the postsynapse.

Reply: We thank the referee for pointing this out. We changed the sentence.

“Physiological Tau locates to axons where it associates with microtubules.” This statement implies tau is an axon-specific protein and must be changed to reflect that tau is located throughout the neuron where it binds microtubules.

Reply: Thanks for pointing this out. We changed the sentence to “*Tau is located throughout the neuron and regulates the dynamics and organization of microtubules*”.

- “In Alzheimer’s disease, phosphorylation of Tau by the tyrosine kinase Fyn dissociates Tau from the microtubules resulting in Tau missorting to the postsynapse (Fig. 4A).” This statement is false, this has not been shown in Alzheimer’s disease.

Reply: We thank the referee for raising this issue. We removed this sentence.

- “The data suggest that phosphorylation and other post-translational modifications may predominantly promote detachment of Tau from microtubules and subsequent redistribution of Tau from axons to dendritic spines.” The data in this manuscript do not speak to these points at all. Also, tau does not detach from microtubules, some specific modifications can reduce binding affinity. The unsubstantiated dogma of tau redistribution is noted above.

Reply: We thank the referee for pointing out this issue. We removed this sentence.

- The statement that in vitro-formed 4-5 protein condensates are “representing the postsynaptic density (PSD)” is an overstretching assertion.

Reply: We changed this to “*Using biochemical reconstitution, we showed that Tau concentrates inside in vitro condensates and membrane-associated clusters, which are formed by major PSD scaffold proteins and were previously suggested to mimic specific aspects of the postsynaptic density.*”

The studies reported do not clearly rule out the possibility that tau is impacting one of the other in vitro PSD condensate proteins as the authors have only assessed the impact of tau on PSD95 and N2RB. Without looking at each of the other proteins this remains unclear.

Reply: Thanks for the thoughtful suggestion. We investigated potential interactions of Tau with the other three PSD condensate proteins by NMR spectroscopy. NMR HSQC experiments of ¹⁵N-labelled Tau with either GKAP, Shank, or Homer did not reveal any binding (new Supplementary Fig. 3), as stated in the revised manuscript: “*In contrast, no interaction between Tau and any of the three other PSD scaffold proteins (GKAP, Shank, Homer) was detected by NMR spectroscopy*”. Therefore, we propose that the Tau-mediated arrest of PSD condensates primarily relies on the multivalent Tau/PSD-95 interactions.

Supplementary Fig. 3 | Interactions between Tau and PSD proteins. NMR signal intensity profiles of Tau upon the addition of 6-fold molar excess of PSD-95 (orange bar), GKAP (cyan lines), Shank (green lines) or Homer (purple lines), respectively. I_0 represents the signal intensities of Tau alone. The domain organization of Tau is shown below the intensity profiles. For Tau/GKAP experiments, 20 μ M Tau and 120 μ M GKAP were used. For others, 10 μ M Tau and 60 μ M of PSD-95/Shank/Homer were used due to limited protein stocks.

The pDLS peptide competition studies with NMR and droplet assays are a good initial set of studies showing some degree of domain specificity. However, a set of experiments that more directly demonstrate the specificity of the condensate interaction between tau and PSD95 (and potentially other constituents, per comment above) would strengthen the work. For example, testing deletion or point mutation versions of tau, PSD95, N2RB and other PSD proteins (potentially) that are predicted to interfere with their interaction in condensates would strengthen the work. One would predict that ideally they can phase separate on their own but would not coalesce in the same condensate.

Reply: Thanks for the valuable suggestion. Please note that we have already tested the truncated version of PSD-95 (i.e. PSD-95_SH3-GK; indicating SH3 and GK domain of PSD-95 are sufficient for Tau/PSD-95 interaction) and Fyn-phosphorylated Tau. For the revision of this manuscript, we additionally performed NMR-based PSD-binding experiments with Y394N mutant Tau and c-Abl-phosphorylated Tau. The new data indicate that the Y394N point mutation and c-Abl phosphorylation of Tau do not lead to a detectable change in the overall interaction profiles of Tau and PSD-95 (new Supplementary Fig. 4B). We believe this is because of the highly multivalent nature of the Tau/PSD-95 interaction and the fact that the binding process is on the intermediate NMR time scale. Binding processes on the intermediate NMR time scale lead predominantly to peak intensity attenuation and thus can make it difficult to detect changes in the interaction that might only slightly affect the kinetics of exchange between the bound and free forms of Tau.

We also tried to purify a PSD-95_SH3 construct to test whether the SH3 domain alone can interact with Tau. However, the PSD-95_SH3 construct displayed very low stability due to the lack of the GK domain in agreement with a previous report.¹

1. McGee, Aaron W., et al. "Structure of the SH3-guanylate kinase module from PSD-95 suggests a mechanism for regulated assembly of MAGUK scaffolding proteins." *Molecular cell* 8.6 (2001): 1291-1301.

Supplementary Fig. 4 | Tyrosine phosphorylation of Tau and its impact on NMR intensity profiles. **A, B.** Superposition of ^1H - ^{15}N HSQC spectra of unmodified Tau (green) with Fyn-phosphorylated Tau (blue; pTau^{Fyn}) in (A) and c-Abl-phosphorylated Tau (purple; pTau^{c-Abl}) in (B). The tyrosine residues being phosphorylated (highlighted in bold) and their neighboring residues are shown in the zoom-in views. **C.** NMR chemical shift perturbation analysis of Tau upon Fyn-mediated (blue bars) or c-Abl-mediated (purple lines) phosphorylation. **D.** NMR signal intensity profiles showing the signal intensity changes of Tau (orange bars), pTau^{c-Abl} (blue lines), pTau^{Fyn} (dark blue lines), and Tau_Y394N (dark red

lines) peaks upon the addition of 6-fold molar excess of PSD-95, with I_0 being signal intensities of Tau alone, pTau^{c-Abl} alone, pTau^{Fyn} alone and Tau_Y394N alone, respectively. 10 μ M of different Tau constructs and 60 μ M of PSD-95 were used. The domain organization of Tau is illustrated below the intensity plots. In **C** and **D**, grey bars represent the position of Tau's five tyrosine residues.

The authors state that tau had a dose-dependent effect on mobile fractions in Figures 1 and 2, but the data do not support a dose-dependent effect. To make this conclusion, each dose must be significantly different from the others – this does not appear to be the case based on statistics shown.

Reply: We thank the referee for the suggestion. We removed the term “dose-dependent”.

The authors did not mention whether Y197 was modified in the pTau-Fyn sample. Moreover, if Fyn-mediated phosphorylation is a critical component to tau's role in the PSD then the work would be strengthened by determining the specificity of the effects for each of the 5 available tyrosine residues in tau.

Reply: Thanks for the considerate suggestion. Based on our NMR data shown in supplementary Fig. 4A and 4C, we did not observe significant changes in the Tau peaks around Y197 upon Fyn phosphorylation. This indicates that Y197 was not phosphorylated by Fyn in the chosen conditions.

Additionally, we used the c-Abl-phosphorylated Tau (which phosphorylated Tau at Y18, Y197, Y310 and Y394) and Tau_Y394N to test their binding to PSD-95 by NMR titration experiments. We did not observe significant changes in Tau/PSD-95 interaction profiles upon the Y394N point mutation or c-Abl-mediated Tau phosphorylation. As discussed above, we believe that this is because of the multivalent interaction, i.e. multiple segments in Tau binding to PSD-95, and the intermediate exchange time scale of the binding process. We therefore currently cannot determine the specificity of the five tyrosine residues for the Tau/PSD-95 interaction.

Based upon the methods section, it does not appear that a condensate-inducing crowding agent (e.g. PEG, RNA, etc.) was used in the fluid condensate experiments. Please clarify.

Reply: Yes, this is correct (now clarified in the methods section).

It's striking that all the data are interpreted in the context of pathological tau or toxicity mechanisms. Given the work that was done, it is unclear how the authors have ruled out a that these tau-mediated effects on PSD condensates reflect a potential normal physiological function.

Reply: We thank the referee for encouraging us to interpret the data in the context of the physiological mechanism. There are some studies focusing on the physiological function of dendritic Tau, indicating that Tau is involved in the rearrangement of cytoskeletal fibers, brain-derived neurotrophic factor (BDNF)-induced synaptic plasticity, hippocampal neurogenesis, regulation of interactome of RNA-binding protein TIA1, and stress granule formation¹⁻³. However, compared to the increasing evidence suggesting a pathological gain of function of dendritic Tau⁴⁻⁶, the potential loss of its physiological functions, remains poorly understood and requires further investigation. Therefore, we believe that it is currently difficult to interpret data in the context of physiological function.

1. Chen, Qian, et al. "Tau protein is involved in morphological plasticity in hippocampal neurons in response to BDNF." *Neurochemistry international* 60.3(2012): 233-242.
2. Pallas-Bazarra, Noemí, et al. "Novel function of Tau in regulating the effects of external stimuli on adult hippocampal neurogenesis." *The EMBO journal* 35.13 (2016): 1417-1436.
3. Vanderweyde, Tara, et al. "Interaction of tau with the RNA-binding protein TIA1 regulates tau pathophysiology and toxicity." *Cell reports* 15.7 (2016): 1455-1466.

4. Bi, Mian, et al. "Tau exacerbates excitotoxic brain damage in an animal model of stroke." *Nature communications* 8.1 (2017): 473.
5. Ittner, Lars M., et al. "Dendritic function of tau mediates amyloid- β toxicity in Alzheimer's disease mouse models." *Cell* 142.3 (2010): 387-397.
6. Schaler, Ari W., et al. "PAC1 receptor-mediated clearance of tau in postsynaptic compartments attenuates tau pathology in mouse brain." *Science translational medicine* 13.595 (2021): eaba7394.

There doesn't appear to be any validation of recombinant proteins

Reply: Thanks for pointing this out. We now provide a gel picture showing the purity of our recombinant proteins (Supplementary Fig. 5). Please note that Tau has anomalous mobility on SDS-PAGE, appearing at the molecular weight of around 60 kDa instead of the expected 45.8 kDa ¹.

1. Danis, Clément, et al. "Nuclear magnetic resonance spectroscopy for the identification of multiple phosphorylations of intrinsically disordered proteins." *JoVE (Journal of Visualized Experiments)* 118 (2016): e55001.

In the revised version of the manuscript we state: "Protein purity was analyzed by running 4%-20% precast polyacrylamide gel (BIO-RAD) in the standard SDS-containing Tris-Glycine buffer, followed by Coomassie blue staining (Supplementary Fig. 5)."

Supplementary Fig. 5 | Gel-based analysis of purified proteins. **A.** 2 μ g of each protein was loaded into the well of 4%-20% SDS-polyacrylamide gel. **B.** Molecular weights of the proteins calculated by the online Protparam tool (<https://web.expasy.org/protparam/>). His-NR2B stands for His₆-tagged GCN4-NR2B; PSD-95 and Homer stand for full-length PSD-95 and Homer, respectively; GKAP stands for the GKAP_GBR_PBM construct; Shank stands for Shank_M1718E_PDZ_HBS_CBS_SAM construct; Tau stands for full-length human 2N4R Tau.

Figure 1 legend: "The Alzheimer's disease-associated protein Tau...". This is an inappropriate way to talk about tau protein.

Reply: We tried to tune down such statements in the revised version of the manuscript, but believe that there is very strong experimental evidence that tau is associated in some ways with Alzheimer's disease.

Figure 1C – it is not clear how it was determined that the arrows are indicating/depicting fusing droplets.

Reply: Thanks for encouraging us to clarify this. The fusion of the PSD condensates occurs rapidly, making it difficult to capture a video. Therefore, we have included images depicting the droplets undergoing coalescence into larger ones. This method of indication was also used in a previous study on the phase separation of PSD proteins ¹.

1. Zeng, Menglong, et al. "Reconstituted postsynaptic density as a molecular platform for understanding synapse formation and plasticity." *Cell* 174.5 (2018): 1172-1187.

As noted above, the suggested distribution of tau in Figure 4A is inaccurate and the notion of tau mislocalization being responsible for somatodendritic tau (in disease) is not supported by evidence in human disease and is based largely on the assumption it is not normally in the somatodendritic compartment and data from cultured neurons.

Reply: Thanks for identifying the issue. We removed this panel from Fig. 4.

Figure 4B – it is not Y187 it is Y197.

Reply: Thanks for noting the typo. We corrected it in Fig. 4A (i.e. previous Figure 4B; previous Figure 4A – the cartoon for Tau mislocalization was removed).

Figure 5. The model stretches the data in this work too far. The study was focused on tau altering the dynamics of a small set of PSD-associated proteins using in vitro phase separation experiments. This does not inform the role of microtubule binding, effects of fyn phosphorylation on microtubule binding, and synaptic functional changes.

Reply: Thanks for raising the issue. We significantly revised the figure and only kept the part indicating that Tau alters the dynamics of PSD condensates/clusters.

Fig. 5 | Proposed model illustrating Tau-mediated dynamic arrest of the PSD drives synaptic dysfunction. At the postsynapse, Tau/pTau^{Fyn} concentrate in the PSD, bind to PSD-95 through multivalent interactions, and decrease PSD dynamics. Dynamically arrested PSD condensates/clusters enhance the clustering and retention time of NMDA receptors on the membrane.

Reviewer #3:

The manuscript by Shen et al. describes the interaction of tau with Post-synaptic density proteins in vitro and the formation of condensates. In the presence of tau, the recovery after photobleaching is reduced and this is even more so in the presence of Fyn phosphorylated tau. The authors nicely

demonstrate that the reduced recovery is partially rescued in the presence of a design peptide which competes for the supposed binding site of PSD95. The experimental work is then very nicely supported by NMR studies to investigate interactions between the molecular components. Overall, this is an interesting piece of work that provides insight into the mechanisms by which tau may influence the properties of the PSD resulting in alteration in NMDA receptor activity. The paper ends with a suggested mechanism that relates the increase in phosphorylation of tau with its reduced binding to microtubules and mislocalisation to the PSD.

Reply: We are grateful for the careful review and highlighting the importance of our work.

Importantly, I believe the authors may want to consider the different isoforms of tau that are present in the cell and it is not possible to assume that all are the same. It may be that the tau that associates with microtubules differs from the one that associates with PSD and there is nothing in this paper that can provide further insights into this. Therefore, some of the "definite" language regarding temporal processes (e.g when something is phosphorylated and where it goes) is speculative. This does not diminish the impressive work provided here but merely points to the care needed when referred to "tau" as a general term.

Reply: Thanks for the helpful suggestion. Combined with other referees' suggestions, we removed many speculative statements, especially those regarding the Tau/microtubule association and Tau mislocalization to the postsynaptic site. As the referees indicated in the comments, we believe that the impact of our work does not depend on the occurrence of Tau mislocalization to the postsynaptic site.

Furthermore, for this revision, we employed the 2N3R Tau isoform in the NMR titration experiments with PSD-95. The overall interaction profiles (except the missing R2 region in 2N3R) between PSD-95 and 2N3R Tau do not exhibit substantial differences compared to those observed with 2N4R Tau.

Figure legend: NMR signal intensity plots presenting the signal intensity changes of 2N4R Tau (orange lines) and 2N3R Tau (grey bars) HSQC peaks upon the addition of 6-fold molar excess of PSD-95, with I_0 being signal intensities of 2N4R Tau alone and 2N3R Tau alone, respectively. The domain organizations of 2N4R Tau and 2N3R Tau are shown below the intensity plots; N1 and N2 stand for the N-terminal inserts; PRD stands for the proline-rich domain; R1, R2, R3, and R4 represent the microtubule-binding repeats of Tau. 20 μM of 2N4R/2N3R Tau and 120 μM of PSD-95 were used in the experiments.

We decided to not include these data into the manuscript, because many additional experiments would be required to thoroughly compare 4R and 3R Tau in the context of the PSD.

Also, the work provided here is in vitro and is extremely nicely performed at relatively low concentrations. However, it is not possible to definitively say this is what is happening in the context of a cellular milieu so again, I would advise some rephrasing to account for this.

Reply: We thank the referee for the valuable suggestion. We carefully revised all parts of the manuscript (including title and abstract) to clarify that the work reported focuses on in vitro studies.

I have some Comments which really refer to the early part of the paper:

1. The authors mention in the introduction that phosphorylation occurs prior to aggregation. However, it is far from determined exactly when phosphorylation occurs on tau and although they refer to three papers, they could easily find more that state the opposite view. It is not clear which tau isoform they are referring to and it is not certain whether all tau aggregates, and what the temporal organisation of the process may be. It may be that the tau that associates in PSD never forms aggregates.

Reply: Thanks for the thoughtful suggestion. We removed this sentence to avoid misleading.

2. Results: The type of tau being used should be stated here when it is first introduced (without requiring reference to the methods)

Reply: Thanks for raising the issue. We indicated the type of Tau used throughout the project (i.e. full-length 2N4R Tau) at the beginning of the result part.

3. Figure 1B shows droplets for the mixture of the PSD proteins and then the addition of tau also shows droplets. It is not entirely clear to me what the authors mean when they say the four components of PSD form LLPS, and then the tau leads to spherical droplets. What is the difference. The authors say that larger droplets are formed in the presence of tau (1C) but there is no quantification and a single image is provided. Either this observation should be removed, or justified with quantification.

Reply: Thank you for asking. The formation of spherical droplets is a key feature of the phase separation phenomenon. Here, we tried to describe “mixing four components of PSD” (i.e. PSD 4X in the upper part of Fig. 1B) and “mixing four components of PSD and Tau” (i.e. PSD 4X + Tau in the bottom part of Fig. 1B) both can result in the phase separation and formation of phase-separated spherical droplets. The quantification of the number and size of phase-separated droplets formed are shown in Fig. 4E and 4F.

To enhance clarity in our manuscript, we have revised the sentences as follows: “*Mixing of the four PSD scaffold proteins resulted in immediate phase separation, in agreement with previous observations (Fig. 1B; upper part). We then mixed the four PSD scaffold proteins and full-length human 2N4R Tau (441 residues; further referred to as Tau) at an equal molar ratio in the same buffer and observed the formation of phase-separated spherical droplets (Fig. 1B; bottom part).*”

4. 1H shows curves of FRAP and it is interesting to note very little difference between 0 and 0.5 tau. The curves overlap and even cross. In 1G, there appears to be a "significant" difference. In the legend, it states that 4-6 curves are included for FRAP but the number of images used for 1G is not included. Why would these show different differences?

Reply: Thanks for the inquiry. Fig. 1H shows the averaged FRAP curves, while Fig. 1G shows the representative image from each FRAP experiment. The mobile fractions were plotted in Fig. 1I.

We think the referee might ask about the different differences between the FRAP curves in 1H and mobile fractions in 1I, so here we explained how we calculated the mobile fractions:

As explained in the method part, we fitted the averaged FRAP curve with a bi-exponential equation and calculate the mobile fraction from the fitted curve, based on the following equation:

$$\text{Mobile fraction} = \frac{I_{\text{plateau}} - I_i}{I_{\text{prebleach}} - I_i}$$

We then normalized the intensity of the frame before bleaching ($I_{\text{Prebleach}}$) as 100% and just after bleaching ($I_{\text{Postbleach}}$ at time 0; called I_i) as 0%. In this context, the mobile fraction represents the value of the I_{plateau} , which corresponds to the point where the FRAP curve reaches a plateau. However, it is important to note that the FRAP curves may not reach the plateau within the 260-second recording period. As a result, the endpoint values of the FRAP curves at 260 seconds (shown in Fig. 1G) do not necessarily match the mobile fractions (shown in Fig. 1I) calculated from the I_{plateau} .

5. As a non-NMR specialist this could be a naive question. Can the authors see all 441 residues in the NMR spectra? I ask this because I know that purification of 2N4R is notoriously difficult and gives rise to truncations and fragmentation of the protein. It might have been nice to show some data on the purity of the tau.

Reply: Thanks for the suggestion. We cannot see all 441 residues in the NMR HSQC spectra. For example, we cannot see any proline residues in the HSQC spectra. For the concern regarding Tau purity, we now provide a gel picture illustrating the purity of our recombinant proteins (Supplementary Fig. 5). Please note that Tau has anomalous mobility on SDS-PAGE, appearing at the molecular weight of around 60 kDa instead of the expected 45.8 kDa ¹.

1. Danis, Clément, et al. "Nuclear magnetic resonance spectroscopy for the identification of multiple phosphorylations of intrinsically disordered proteins." *JoVE (Journal of Visualized Experiments)* 118 (2016): e55001.

We describe this now in the methods part: "Protein purity was analyzed by running 4%-20% precast polyacrylamide gel (BIO-RAD) in the standard SDS-containing Tris-Glycine buffer, followed by Coomassie blue staining (Supplementary Fig. 5)."

Supplementary Fig. 5 | Gel-based analysis of purified proteins. A. 2 µg of each protein was loaded into the well of 4%-20% SDS-polyacrylamide gel. **B.** The molecular weight of each protein was calculated by the online ProtParam tool (<https://web.expasy.org/protparam/>). His-NR2B stands for His₆-tagged GCN4-NR2B; PSD-95 and Homer stand for full-length PSD-95 and Homer, respectively; GKAP stands for the GKAP_GBR_PBM construct; Shank stands for Shank_M1718E_PDZ_HBS_CBS_SAM construct; Tau stands for full-length human 2N4R Tau.

6. In the discussion the authors state "The data suggest that phosphorylation and other post-translational modifications may predominantly promote detachment of Tau from microtubules and subsequent redistribution of Tau from axons to dendritic spines 11,12". Here they do refer to literature but the sentence suggests they have data that relates to this here. They don't and this sentence requires rewording so it is clear this is related to literature findings not those included in this paper.

Reply: Thanks for the thoughtful suggestion. We removed this sentence to avoid misleading.

MINOR

1. Abstract: I think arrest should have an s on the end (depending on whether TAU is a plural or singular term).

Reply: We thank the referee for noting the problem. Combined with the other referee's suggestion, we changed the title to "*Multivalent Tau/PSD-95 interactions arrest in vitro condensates and clusters mimicking the postsynaptic density*".

Reviewer #4:

This article investigates a possible mechanism by which the protein Tau can cause synaptic dysfunction in Alzheimer's disease (AD) prior to neuronal death. The premise of this work is that Tau, which is usually associated with microtubuli, gets missorted in AD to the postdynamapse. Here, the authors determine possible effects of Tau on protein condensates, particularly on PSD condensates, which are known to form at the postdynamapse and organize the location and dynamics of the NMDA receptor. They show that Tau co-condensates with the PSD proteins and that the presence of Tau decreases the mobility of these condensates. Phosphorylated Tau increases this effect. They further determine the binding interface of Tau with PSD-95 and show that a PSD-95 peptide can competitively reverse the effects of Tau on the condensates. This thoroughly conducted study tackles the important and timely question if proteins involved in AD can have an effect on neuronal protein condensate formation. I recommend its publication after a few concerns have been addressed.

Reply: We thank the referee for the careful evaluation of our manuscript and for highlighting the importance of the work.

My major concern is that the manuscript does not show a direct effect of Tau on NMDA receptor mobility (Fig. S2). This is by itself not a problem but the discussion section later suggests that the paper did show "Tau-induced decrease in NMDA receptor mobility" because "reduced PSD-95 dynamics results in decreased NMDA receptor mobility." (Page 12). This is not necessarily the case, in fact the authors show that PSD-95 has a much higher mobility in the NR2B/PSD condensates than NR2B. Therefore, the data rather suggest that Tau affects the PSD portion of the NR2B/PSD condensates but not the NR2B construct.

Reply: Thanks for the suggestion. In our discussion, the "Tau-induced decrease in NMDA receptor mobility" statement was motivated by our observation of decreased dynamics of the NMDA

truncated construct (which contains the last five amino acids of the NMDA receptor subunit NR2B fused to the C-terminus of the tetrameric GCN4 coiled-coil domain) upon the addition of Tau in fluid condensates (Figure 1G-1I). However, as pointed out by the referee, in membrane-bound condensates, "PSD-95 has a much higher mobility in the NR2B/PSD condensates than NR2B." Therefore, we believe that Tau directly and predominantly affects PSD-95 dynamics in NR2B/PSD condensates via PSD-95/Tau multivalent interactions (Figure 3) and may only indirectly influence NMDA receptor dynamics through the PSD-95/NMDA receptor interaction.

To avoid misunderstanding, we now state in the discussion: *"Additionally, the Tau-mediated arrest decreased the solution condensate dynamics of a truncated NMDA construct, which can bind to PSD-95."*, as well as *"We demonstrated that Tau concentrates in PSD condensates and membrane-anchored clusters, and decreases the mobility of PSD-95. Our results propose a relay mechanism in which multivalent interactions between the intrinsically disordered protein Tau and the GK domain of PSD-95 causes cross-linking of PSD-95 molecules, leading to dynamic arrest of PSD condensates and clusters and thus indirectly to decreased NMDA receptor mobility through PSD-95 and NMDA receptor interaction."*.

Minor comments

Page 2: "an electron-dense molecular cluster" does this refer to EM images? Unclear in context.

Reply: Thanks for the suggestion. We rephrased it: *"In electron micrographs of excitatory synapses"*.

Page 6: "2N4R Tau" which Tau constructs were used in the previous experiments?

Reply: Thanks for encouraging us to clarify this. Throughout the project, we used the longest human Tau isoform (2N4R with 441 amino acids). We highlighted that we used 2N4R Tau when first introduced in the result part: *"We then mixed the four PSD scaffold proteins and full-length human 2N4R Tau (441 residues; further referred to as Tau) at an equal molar ratio in the same buffer and observed the formation of phase-separated spherical droplets (Fig. 1B; bottom part)."*

Fig. 3C discussed on page 7: Because I₀ is buried under the X-bar and not visible at all, it is a bit strange to give it its own legend (same with 3F). The text on page 7 talks interchangeably of line intensity (which is measured) and line broadening (which is not measured).

Reply: Thanks for the careful review. We removed the I₀ legends. Also, we replaced the term "broadening" with *"peak intensity attenuation"*.

Page 9: "does not require binding of Fyn to Tau" unclear.

Reply: We thank the referee for noting this. We removed this sentence to avoid misleading.

Page 11: "The data suggest that phosphorylation and other post-translational modifications may promote detachment of Tau from microtubules and subsequent redistribution of Tau from axons to dendritic spines" What data are referred to? The data in the paper do not show any of this.

Reply: We thank the referee for pointing this out. We removed this sentence to avoid misleading.

Reviewer #1 (Remarks to the Author):

The revised paper contains significant textual changes but few additional data. No data that address the central issue with this paper were added. This issue is that all experiments are in vitro experiments whose relevance to in vivo functions of PSD95 and tau remains untested. Especially since tau deletions have no postsynaptic phenotype that has been reported, I suspect that the authors are studying an in vitro phenomenon whose biological significance is questionable. Without data demonstrating that the in vitro processes the authors study have a role in an actual synapse I feel unable to recommend this paper for publication.

Reviewer #2 (Remarks to the Author):

Considering the lack of in-cell data and remaining uncertainty of the physiological relevance of these findings it is critical that the discussion includes some comments addressing the caveats and/or limitations of the approaches/model used.

In the prior review the following comment was made: It's striking that all the data are interpreted in the context of pathological tau or toxicity mechanisms. Given the work that was done, it is unclear how the authors have ruled out a that these tau-mediated effects on PSD condensates reflect a potential normal physiological function.

The author's reply was: We thank the referee for encouraging us to interpret the data in the context of the physiological mechanism. There are some studies focusing on the physiological function of dendritic Tau, indicating that Tau is involved in the rearrangement of cytoskeletal fibers, brain-derived neurotrophic factor (BDNF)-induced synaptic plasticity, hippocampal neurogenesis, regulation of interactome of RNA-binding protein TIA1, and stress granule formation 1-3. However, compared to the increasing evidence suggesting a pathological gain of function of dendritic Tau 4-6, the potential loss of its physiological functions, remains poorly understood and requires further investigation. Therefore, we believe that it is currently difficult to interpret data in the context of physiological function.

Perhaps it is fair to suggest the physiological function of tau in synapses is still unclear, however, multiple studies have shown evidence of synaptic tau in control/physiological conditions (from human tissue studies to model-based studies) (for example: Ittner et al. Cell, 2010; Schaler et al. JTM, 2021; Tai et al. AJP, 2012; Frandemiche et al. JN 2014; Mondragón-Rodríguez et al. JBC 2012). In addition, 3 out of the 4 main figures in the main paper use "physiological" tau proteins for the experiments and yet the interpretation of the work is under the lens of pathological events. It is not clear where the link to disease comes from in regard to data in figures 1-3, which drove my initial comment that the authors should consider discussing what their data suggests for potential physiological roles of tau in the synapse. One cannot disregard the reported presence of tau in synapses (or synaptosomal preps) in WT or control conditions because a few more prior studies focused on pathological contexts (although at least two of the references provided that show pathological synaptic tau also clearly show there is synaptic tau in control conditions, Ittner and Shaler papers), and the presumption is that these data would have relevance to the potential functions of that pool of tau.

Fig 1B,C – in the figure it says 10 uM but in the legend says 20 uM. Arrows to show fusing droplets are not convincing. It is not that it is difficult to believe that fusion occurs in these conditions (as would be expected), it is that the images do not convey what the authors want them to convey. Time-lapse images or even videos would better illustrate droplet fusion.

Figure 1I – only differences to no tau are displayed, were no other groups different from one another? Please clarify.

Figure 2E – only differences to no tau are displayed, were no other groups different from one another? Please clarify.

Figure 4B-F is a little confusing. Is E and F quantification of the droplets in B and C? Also, images

supporting the quantification of droplet size and number should be added for all groups.
Figure 4B,C – Indicate tau amount as in Fig 1. Arrows to show fusing droplets is not convincing (see above comment).
Figure 4G – all conditions in the quantitative data in H and I should be shown in G.

Reviewer #3 (Remarks to the Author):

The manuscript is improved following reviewers comments and in particular, the in vitro nature of the work is now clearer. I am satisfied by the response and amendments to reflect this. I do suggest that the nature of the tau is highlighted. I understand that the authors use "tau" as notation for 2N4R tau, but I think at least when introducing a new form - for example the phosphorylated form- it is helpful to reiterate that this is being used to produce phospho-tau using Fyn.

Overall, I think the work is nicely conducted. It would be great to see whether these interactions are clarified in a cellular setting This work provides a great stepping stone towards this.

Reviewer #4 (Remarks to the Author):

The authors addressed all issues brought up in my previously review. Therefore, I recommend the publication of this article.

We extend our gratitude once more to all the referees for their careful assessment of our revised manuscript and their positive feedback on the improvements we implemented. We thoroughly reviewed the new comments, and, as indicated below, we have provided our responses in blue text.

Reviewer #1:

The revised paper contains significant textual changes but few additional data. No data that address the central issue with this paper were added. This issue is that all experiments are in vitro experiments whose relevance to in vivo functions of PSD95 and tau remains untested. Especially since tau deletions have no postsynaptic phenotype that has been reported, I suspect that the authors are studying an in vitro phenomenon whose biological significance is questionable. Without data demonstrating that the in vitro processes the authors study have a role in an actual synapse I feel unable to recommend this paper for publication.

Reply: Thanks for the suggestion. We added a few sentences to the discussion section stating that *in vivo* experiments are required to further understand and characterize the role of tau in the PSD and to better understand the molecular processes driving the formation of the PSD:

“Therefore, future in vivo experiments and approaches are critical for understanding the molecular process driving PSD formation, characterizing Tau/PSD-95/NMDAR interaction within PSDs, and clarifying the physiological and pathological roles of Tau in PSD regulation.”

Reviewer #2:

Considering the lack of in-cell data and remaining uncertainty of the physiological relevance of these findings it is critical that the discussion includes some comments addressing the caveats and/or limitations of the approaches/model used.

Reply: Thanks for the suggestion. We added a few sentences to the discussion section stating that *in vivo* experiments are required to further understand and characterize the role of tau in the PSD and to better understand the molecular processes driving the formation of the PSD:

“Therefore, future in vivo experiments and approaches are critical for understanding the molecular process driving PSD formation, characterizing Tau/PSD-95/NMDAR interaction within PSDs, and clarifying the physiological and pathological roles of Tau in PSD regulation.”

In the prior review the following comment was made: It’s striking that all the data are interpreted in the context of pathological tau or toxicity mechanisms. Given the work that was done, it is unclear how the authors have ruled out a that these tau-mediated effects on PSD condensates reflect a potential normal physiological function.

The author’s reply was: We thank the referee for encouraging us to interpret the data in the context of the physiological mechanism. There are some studies focusing on the physiological function of dendritic Tau, indicating that Tau is involved in the rearrangement of cytoskeletal fibers, brain-derived neurotrophic factor (BDNF)-induced synaptic plasticity, hippocampal neurogenesis, regulation of interactome of RNA-binding protein TIA1, and stress granule formation 1-3. However,

compared to the increasing evidence suggesting a pathological gain of function of dendritic Tau 4-6, the potential loss of its physiological functions, remains poorly understood and requires further investigation. Therefore, we believe that it is currently difficult to interpret data in the context of physiological function.

Perhaps it is fair to suggest the physiological function of tau in synapses is still unclear, however, multiple studies have shown evidence of synaptic tau in control/physiological conditions (from human tissue studies to model-based studies) (for example: Ittner et al. Cell, 2010; Schaler et al. JTM, 2021; Tai et al. AJP, 2012; Frandemiche et al. JN 2014; Mondragón-Rodríguez et al. JBC 2012). In addition, 3 out of the 4 main figures in the main paper use “physiological” tau proteins for the experiments and yet the interpretation of the work is under the lens of pathological events. It is not clear where the link to disease comes from in regard to data in figures 1-3, which drove my initial comment that the authors should consider discussing what their data suggests for potential physiological roles of tau in the synapse. One cannot disregard the reported presence of tau in synapses (or synaptosomal preps) in WT or control conditions because a few more prior studies focused on pathological contexts (although at least two of the references provided that show pathological synaptic tau also clearly show there is synaptic tau in control conditions, Ittner and Shaler papers), and the presumption is that these data would have relevance to the potential functions of that pool of tau.

Reply: Thanks for the suggestion. We extended the discussion to highlight the potential physiological activities of Tau in the post-synapse:

“On the other hand, recent studies have increasingly demonstrated the presence of synaptic Tau in physiological conditions and its various physiological functions within the synapse, including brain-derived neurotrophic factor (BDNF)-induced synaptic plasticity, hippocampal neurogenesis and long-term depression. Our study demonstrates that wild-type 2N4R Tau can decrease the PSD dynamics, suggesting a possible physiological role of Tau in stabilizing the major PSD scaffold protein PSD-95 and facilitating the downstream signaling pathways of NMDA receptors.”

Fig 1B,C – in the figure it says 10 uM but in the legend says 20 uM. Arrows to show fusing droplets are not convincing. It is not that it is difficult to believe that fusion occurs in these conditions (as would be expected), it is that the images do not convey what the authors want them to convey. Time-lapse images or even videos would better illustrate droplet fusion.

Reply: Thanks for noting the typo, we corrected it in the figure legend. We also removed the arrows to avoid misleading.

Figure 1I – only differences to no tau are displayed, were no other groups different from one another? Please clarify.

Reply: Thanks for asking to clarify this. We checked all potential multiple comparisons across various conditions, as illustrated in the image below. With the exception of the comparisons between "0.5 μ M Tau" and "1 μ M Tau" as well as "10 μ M Tau" and "20 μ M Tau", significant differences were observed among the remaining groups.

We consider the comparisons to the "0 μ M Tau" (i.e. no tau) condition to be the most important one, highlighting that the addition of Tau decreases the dynamics of NMDA receptor peptide. Hence, we only displayed those comparisons with the "0 μ M Tau" condition in the main figure.

Figure 2E – only differences to no tau are displayed, were no other groups different from one another? Please clarify.

Reply: Thanks for the inquiry. As illustrated below, there were no other groups different from one another.

Figure 4B-F is a little confusing. Is E and F quantification of the droplets in B and C? Also, images supporting the quantification of droplet size and number should be added for all groups.

Reply: Thanks for the comment. The images supporting the number and size quantification of "PSD 4X" and "PSD 4X + Tau" have already been shown in Figures 1B and C. The images supporting the number and size quantification of "PSD 4X + pTau^{Fyn}" have been shown in Figures 4B and C. We made these clearer in the figure legend of Figures 4E and F:

"For (E) and (F), The images corresponding to PSD droplets alone and with Tau are shown in Fig 1B and 1C; the images of PSD droplets with pTau^{Fyn} are shown in Fig 4B and 4C."

Figure 4B,C – Indicate tau amount as in Fig 1. Arrows to show fusing droplets is not convincing (see above comment).

Reply: Thanks for the comment. We add the tau amount to Figures 4B and C. We removed the arrows to avoid misinterpretation.

Figure 4G – all conditions in the quantitative data in H and I should be shown in G.

Reply: Thanks for the suggestion. The images of “PSD 4X” and “PSD 4X + Tau” have been already shown in Fig 1D. We describe this clearer in the figure legend now:

“The corresponding FRAP images of PSD-95 dynamics without and with Tau are shown in Fig 1D.”

Reviewer #3:

The manuscript is improved following reviewers comments and in particular, the in vitro nature of the work is now clearer. I am satisfied by the response and amendments to reflect this. I do suggest that the nature of the tau is highlighted. I understand that the authors use "tau" as notation for 2N4R tau, but I think at least when introducing a new form - for example the phosphorylated form- it is helpful to reiterate that this is being used to produce phospho-tau using Fyn.

Reply: Thanks for the positive feedback and useful suggestion. We added the notation for 2N4R Tau.

Overall, I think the work is nicely conducted. It would be great to see whether these interactions are clarified in a cellular setting This work provides a great stepping stone towards this.

Reviewer #4:

The authors addressed all issues brought up in my previously review. Therefore, I recommend the publication of this article.

Reply: Thanks again for the evaluation and support.